# ELVES: Extraction of Latent Variables with Enhanced Specificity for High-Dimensional Few-Sample Feature Selection

## Abstract

Feature selection for high-dimensional, few-sample data has been a serious issue due to overfitting, high computational complexity and feature redundancy. Here, one key challenge is how to capture characterization of specificity that enhance the outcomes. To tackle this issue, our work proposes a novel supervised feature selection method named ELVES, which exploits the manifold structure of the feature space. Specifically, our method constructs a feature association kernel for each class to capture inter-feature dependencies. By integrating product manifold theory with spectral graph analysis, we develop structure operators that characterize the intrinsic geometry of each class manifold. A graph filtering operator is then designed to produce a filtered operator, whose leading eigenvectors capture class-specific latent variables. These latent variables are iteratively extracted and used to define a feature scoring mechanism that identifies features with strong discriminative power in high-dimensional, few-sample scenarios. Comprehensive experiments demonstrate that ELVES not only improves generalization performance and robustness to few sample size over leading baselines, but also provides new insights into the underlying sources of data variation.

## 1 Introduction

High-dimensional, few-sample data is an increasingly common problem faced in the development of models in natural language processing (Zhang et al., 2022; Brown et al., 2020) and computer vision (Snell et al., 2017; Roy et al., 2022), which arises directly from a wide range of application scenarios such as medical and clinical research (Esteva et al., 2021; Gidwani et al., 2022; Popov et al., 2024), bioinformatics (Hao et al., 2021; Lall et al., 2022), chemometrics (Salaroli & del Carmen Pardo, 2023). Good feature selection methods enhance model accuracy, reduce experimental costs, and, more importantly, assist in understanding the process of data generation (Bolón-Canedo et al., 2022). Existing feature selection methods can be broadly classified into three categories: wrapper, embedded, and filter methods. Wrapper methods require model training on every candidate feature subset that incurs prohibitive computational costs in high-dimensional settings (Alelyani et al., 2018; Roy et al., 2015). Embedded methods integrate feature selection into model training to remove this burden, but they remain susceptible to model-specific bias (Yamada et al., 2020; Cohen et al., 2023). Filter methods rank features independently of any learning model, relying only on statistical or geometric criteria (Cohen et al., 2023), thereby offering high computational efficiency and making them well-suited for high-dimensional, few-sample data.

Existing filter methods fall into three main categories: (i) Statistical metrics based: including ANOVA F-value (Kao & Green, 2008), Fisher score (Duda et al., 2006), information gain (IG) (Vergara & Estévez, 2014), and Gini-index (Shang et al., 2007), each of which independently evaluates the statistical association between a single feature and the target label. (ii) Geometry-based: including Laplacian Score (He et al., 2005; Lindenbaum et al., 2021), SPEC (Zhao & Liu, 2007), and Relief (Kira & Rendell, 1992; Robnik-Šikonja & Kononenko, 2003), all of which assess feature importance by exploiting the geometric structure of the sample space. (iii) Manifold-based: including ManiFeSt (Cohen et al., 2023) and MMDUFS (Yang et al., 2023). By constructing a composite kernel, ManiFeSt is limited to capturing only the shared discriminative variables between classes and cannot extract class-specific latent variables that reveal unique intra-class variation patterns. MM-

DUFS construct a shared structure operator and a differential graph operator across modalities under the product manifold assumption, thereby proposing a multi-modal unsupervised feature selection framework.

It is critical to adequately capture the inter-feature interactions, which is not well done in current filter methods, with most focusing on univariate evaluation, or assessing global associations through sample-point geometry or manifolds (Izetta et al., 2017). Especially in high-dimensional, few-sample scenarios, univariate metrics are susceptible to noise interference, and geometric methods ignore the discriminative information brought about by multi-feature synergy (Cohen et al., 2023). In fact, feature interactions exhibit robustness to few-sample regimes, as they rely on the high-dimensional feature space rather than the limited sample space (Cohen et al., 2023). Consequently, a novel approach grounded in the underlying geometric structure of the feature space holds promise for more accurately uncovering discriminative features.

For high-dimensional, few-sample scenarios, we propose a novel filter-based feature selection method, termed **E**xtraction of **L**atent **V**ariables with **E**nhanced **S**pecificity (ELVES), which leverages the manifold structure of the feature space. Our approach begins by constructing feature association kernels for each class to capture inter-feature dependencies. From these kernels, we derive structure operators that characterize the manifold geometry of each class and compute their symmetric normalized Laplacian matrices. Based on these Laplacians, we then design a graph filtering operator that transforms the structure operators into the filtered operators whose leading eigenvectors (termed differential vectors) can capture the class-specific latent variables of each class. Finally, by iteratively extracting these class-specific latent variables, we define a feature score that can identify the features with high discriminative capability in high-dimensional and few-sample scenarios.

Our contributions are summarized as follows:

1. We propose a novel supervised feature selection method, which explicitly models multivariate feature interactions and captures class-specific latent variables representing underlying differential structure to mitigate feature redundancy and noise interference in high-dimensional and few-sample scenarios.

2. ELVES is first to integrate product manifold constructs with spectral graph analysis for feature-space manifold learning. We provide an asymptotic convergence analysis demonstrating its ability to reliably discover intrinsic difference patterns, thus offering a novel perspective on the sources of data variation.

3. We conduct comprehensive experiments on diverse benchmark datasets to show that ELVES consistently outperforms state-of-the-art baselines, especially enhancing the generalization performance and exhibiting strong robustness to few sample size.

## 2 PROBLEM SETTING AND PRELIMINARY

High-dimensional and few-sample datasets pose a formidable challenge for revealing their underlying distinctive structures. Here, our aim is to extract latent variables that can only be captured by each of class. The principal challenge in dimensionality reduction is to derive a low-dimensional representation of the observation $x_i \in \mathbb{R}^d$ related to the latent variables $\theta \in \mathbb{R}^l$.

For two datasets drawn from different classes within the same modality, their latent spaces may follow a similar underlying structure while also exhibiting unique intra-class structural variations. Consider two datasets $X^{(1)}$ and $X^{(2)}$, each representing a distinct class within a single dataset $X$. Let the observations $x_i^{(1)} \in X^{(1)}$ and $x_i^{(2)} \in X^{(2)}$. Each observation from the dataset can be approximated by the result of a continuous transformation applied to a set of latent variables,

$$(\theta_i, \varphi_i^{(1)}) \xrightarrow{T_1} x_i^{(1)}, \quad (\theta_i, \varphi_i^{(2)}) \xrightarrow{T_2} x_i^{(2)}. \tag{1}$$

The latent variables $\theta$ capture the structure shared by both classes $X^{(1)}$ and $X^{(2)}$, whereas the latent variables $\varphi^{(i)}$ capture the structure specific to $X^{(i)}$.

By extracting these variables, we can identify features associated with the specific structure of each class, thus precisely characterizing the differential patterns and identifying the key discriminative features. To achieve this goal, we use a manifold-based approach. Here, we first present some preliminaries about graph representation and graph signal processing.

## 2.1 Graph Laplacian and Graph Representation

Consider two datasets $X^{(1)} \in \mathbb{R}^{n_1 \times d}$ and $X^{(2)} \in \mathbb{R}^{n_2 \times d}$, where the rows of two matrices correspond to observations from different classes within the same dataset that satisfy the double manifold assumption of Eq. 1. Let $x_i^{(1)}$, $x_i^{(2)}$ denote the $i$-th observation in the $X^{(1)}$ and $X^{(2)}$, respectively. Their affinity matrices $K_{i,j}^{(1)}$, $K_{i,j}^{(2)}$ are computed separately by the following Gaussian kernel functions using the Euclidean norm,

$$K_{i,j}^{(1)} = \exp\left(-\|x_i^{(1)} - x_j^{(1)}\|^2 / 2\sigma_1^2\right), \quad K_{i,j}^{(2)} = \exp\left(-\|x_i^{(2)} - x_j^{(2)}\|^2 / 2\sigma_2^2\right). \tag{2}$$

Here, $\sigma_1$ and $\sigma_2$ are bandwidth parameters that control the decay of each Gaussian kernel.

Let $D^{(1)}$, $D^{(2)}$ be two diagonal matrices whose elements are row sums of $K^{(1)}$ and $K^{(2)}$, respectively. We compute two operators by,

$$Q^{(1)} = (D^{(1)})^{-1/2} K^{(1)} (D^{(1)})^{-1/2}, \quad Q^{(2)} = (D^{(2)})^{-1/2} K^{(2)} (D^{(2)})^{-1/2}. \tag{3}$$

And we denote the symmetric normalized Laplacian matrices by,

$$L^{(1)} = I - Q^{(1)}, \quad L^{(2)} = I - Q^{(2)}. \tag{4}$$

An important property of the Laplacian matrix is that the eigenvectors corresponding to its large eigenvalues can effectively capture the underlying geometric structure of the data. Thus, we extract the differential latent variables $\varphi^{(1)}$, $\varphi^{(2)}$ that can capture the specific structure of $X^{(1)}$ and $X^{(2)}$ respectively by analyzing the two graph Laplacians $L^{(1)}$ and $L^{(2)}$.

## 2.2 Graph Signal Processing and Graph Filters

Consider the signals defined on the vertices of a graph, denoted as real functions $f : V \to \mathbb{R}$. Since the graph has $n$ vertices, these signals can be represented as vectors $f \in \mathbb{R}^n$. The symmetric normalized Laplacian matrix defined in Eq. 4 is positive semi-definite and its eigenvectors form an orthogonal basis of $\mathbb{R}^n$. The Laplacian eigenvectors serve as discrete analogues of the Fourier basis (Ricaud et al., 2019; Shuman et al., 2013). For the symmetric normalized Laplacian, it can be used as a smoothness functional for graph signals (Von Luxburg, 2007). We have

$$f^T L f = \frac{1}{2} \sum_{i,j=1}^{n} K_{i,j} \left( \frac{f_i}{\sqrt{D_{i,i}}} - \frac{f_j}{\sqrt{D_{j,j}}} \right)^2. \tag{5}$$

Combining Eq. 5 with the Courant-Fischer theorem reveals that the eigenvectors corresponding to the smallest eigenvalues minimize the right-hand side under orthonormality constraints (Horn & Johnson, 2012),

$$v_d = \min_{\|f\|_2 = 1;\, f \perp \{v_0, ..., v_{d-1}\}} \frac{1}{2} \sum_{i,j=1}^{n} K_{i,j} \left( \frac{f_i}{\sqrt{D_{i,i}}} - \frac{f_j}{\sqrt{D_{j,j}}} \right)^2. \tag{6}$$

Eq. 6 implies that the eigenvectors corresponding to the small eigenvalues exhibit smoothness in their $\sqrt{D}$-normalized values across neighboring vertices. The Graph Fourier Transform and Graph Inverse Fourier Transform for a signal $f \in \mathbb{R}^n$ are defined as

$$\hat{f}_d = \langle v_d, f \rangle, \quad f = \sum_{d=0}^{n-1} \hat{f}_d v_d. \tag{7}$$

Eq. 7 provides a spectral representation of graph signals. Consequently, the graph spectral filtering can be implemented by eigenvalue-based weights,

$$H(f) = \sum_{d=0}^{n-1} \hat{f}_d h(\lambda_d) v_d. \tag{8}$$

## 3 METHOD

We detail the ELVES proposed for feature selection. It subsequently identifies features associated with differential latent structures that are specific to a single class, and further derives a corresponding feature score.

### 3.1 FEATURE MANIFOLD LEARNING

Consider two datasets $X^{(1)} = [x_1^{(1)}, \cdots, x_d^{(1)}] \in \mathbb{R}^{n_1 \times d}$ and $X^{(2)} = [x_1^{(2)}, \cdots, x_d^{(2)}] \in \mathbb{R}^{n_2 \times d}$, which correspond to two distinct classes derived from a dataset $X \in \mathbb{R}^{n \times d}$ consisting of $n$ samples and $d$ features, where $x_i^{(\ell)} \in \mathbb{R}^{n_\ell}$ denotes the $i$-th feature in the $\ell$-th class, $n_\ell$ denotes the number of samples in the $\ell$-th class, and $n = n_1 + n_2$. To capture class-specific differences in the feature associations, we employ a Gaussian kernel defined in Eq. 2 to learn the underlying geometric structure of the feature space for each class.

Kernel-based approaches are widely employed for nonlinear dimensionality reduction and manifold learning (Roweis & Saul, 2000; Belkin & Niyogi, 2003). In contrast to conventional methods that learn the manifold underlying the samples, our method focuses on the manifold structure of features, thereby capturing multivariate associations. This perspective aligns closely with the framework of graph signal processing.

### 3.2 THE SHARED STRUCTURE OPERATOR

We denote by $V_s$ a matrix whose columns consist of the eigenvectors in both $Q^{(1)}$ and $Q^{(2)}$ defined in Eq. 3 that are associated with shared latent variables $\theta$, and denote by $V_1$ and $V_2$ the matrices whose columns consist of the eigenvectors in $Q^{(1)}$ and $Q^{(2)}$ that are associated with class-specific latent variables $\varphi^{(1)}$ and $\varphi^{(2)}$, respectively. In our ideal setting, the two operators $Q^{(1)}$ and $Q^{(2)}$ can be approximated by,

$$Q^{(1)} \approx V_s V_s^T + V_1 V_1^T, \quad Q^{(2)} \approx V_s V_s^T + V_2 V_2^T. \tag{9}$$

To compute a representation that can capture shared latent structures, we denote by $Q^\theta$ the operator whose columns are equal to the eigenvectors of the symmetric product of $Q^{(1)}$ and $Q^{(2)}$,

$$Q^\theta = Q^{(1)} Q^{(2)} + Q^{(2)} Q^{(1)}. \tag{10}$$

We demonstrate the effectiveness of $Q^\theta$ under the product of manifold setting.

**The Product of manifolds.** Let $\mathcal{M}_a$, $\mathcal{M}_b$ be two manifolds, and let $\mathcal{M} = \mathcal{M}_a \times \mathcal{M}_b$ denote the product manifold. The canonical projection operators $\pi_a : \mathcal{M} \to \mathcal{M}_a$ and $\pi_b : \mathcal{M} \to \mathcal{M}_b$ map a point in $\mathcal{M}$ to its corresponding points in $\mathcal{M}_a$, $\mathcal{M}_b$, respectively. Then we can use the projection operator to extend a real function $f_a : \mathcal{M}_a \to \mathbb{R}$ on $\mathcal{M}_a$ to a function $f : \mathcal{M} \to \mathbb{R}$ over the product $\mathcal{M}$ by $f(x) = f_a(\pi_a(x))$. The datasets $X^{(1)}$ and $X^{(2)}$ contain observations sampled from two product manifolds $\mathcal{M}_1$ and $\mathcal{M}_2$, respectively, where

$$\mathcal{M}_1 = \mathcal{M}_a \times \mathcal{M}_s, \quad \mathcal{M}_2 = \mathcal{M}_b \times \mathcal{M}_s. \tag{11}$$

We assume that the latent variables $\varphi_i^{(1)}$, $\varphi_i^{(2)}$, $\theta_i$ are drawn independently and the observations $x_i^{(1)} \in X^{(1)}$, $x_i^{(2)} \in X^{(2)}$ are computed according to Eq. 1. Let $f_i^{(j)} : \mathcal{M}_j \to \mathbb{R}$ denote the $i$-th eigenfunction of the Laplace-Beltrami operator of the manifold $\mathcal{M}_j$. The eigenfunctions of the products $\mathcal{M}_1$, $\mathcal{M}_2$ are equal to the pointwise product of the eigenfunctions of $\mathcal{M}_a$, $\mathcal{M}_b$, $\mathcal{M}_s$ (Zhang et al., 2021),

$$f_{l,k}^{(1)}(x) = f_l^{(a)}(\pi_a(x)) \cdot f_k^{(s)}(\pi_s(x)), \quad f_{m,k'}^{(2)}(x) = f_m^{(b)}(\pi_b(x)) \cdot f_{k'}^{(s)}(\pi_s(x)). \tag{12}$$

Let $\mu_i^{(j)}$ denote the $i$-th smallest eigenvalue corresponding to the eigenfunction $f_i^{(j)}$. And let $\mu_{l,k}^{(1)}, \mu_{l,k}^{(2)}$ denote the $(l, k)$-th smallest eigenvalue of the products $\mathcal{M}_1$, $\mathcal{M}_2$, where

$$\mu_{l,k}^{(1)} = \mu_l^{(a)} + \mu_k^{(s)}, \quad \mu_{m,k'}^{(2)} = \mu_m^{(b)} + \mu_{k'}^{(s)}. \tag{13}$$

---

**Algorithm 1** Extracting a Single Distinctive Latent Variable

---

1: **Input:** Two datasets $X^{(1)} \in \mathbb{R}^{n_1 \times d}$ and $X^{(2)} \in \mathbb{R}^{n_2 \times d}$, containing $d$ features with $n_1$ and $n_2$ samples respectively. Filter function $h(\lambda) : [0, 1] \to [0, 1]$.
2: **Output:** Differential vectors $\delta^{(1)}, \delta^{(2)} \in \mathbb{R}^d$.
3: Compute the weight matrices $K^{(1)}, K^{(2)}$ via Eq. 2.
4: Compute the operators $Q^{(1)}, Q^{(2)}$ and the symmetric normalized Laplacian matrices $L^{(1)}, L^{(2)}$ via Eq. 3 and 4.
5: Compute the filtering matrices $H(L^{(1)}), H(L^{(2)})$ using Eq. 15.
6: Compute the filtered operators $\tilde{Q}^{(1)}, \tilde{Q}^{(2)}$ using Eq. 16.
7: Compute the differential vectors $\delta^{(1)}, \delta^{(2)}$ from the filtered operators $\tilde{Q}^{(1)}, \tilde{Q}^{(2)}$ respectively.

---

Thus, the matrices $V_s, V_1, V_2$ are mutually orthogonal in pairs (See the Lemma 4 in Appendix F.1). It follows that the operator $Q^\theta$ is equal to,

$$Q^\theta \approx 2V_s V_s^T = \sum_k v_{0,k}^{(1)} (v_{0,k}^{(2)})^T. \tag{14}$$

Therefore, the leading eigenvectors of $Q^\theta$ are associated with the shared latent structure and not the class-specific structures in the product of manifolds.

### 3.3 EXTRACTION OF A SINGLE DISTINCTIVE LATENT VARIABLE

We first compute $Q^{(1)}, Q^{(2)}$ by Eq. 3, and the symmetric normalized Laplacian matrices $L^{(1)}, L^{(2)}$ by Eq. 4. Let $\lambda_i^{(1)}, v_i^{(1)}$ and $\lambda_i^{(2)}, v_i^{(2)}$ denote the leading eigenvalues and eigenvectors of $L^{(1)}, L^{(2)}$, respectively. To extract the class-specific latent variables $\varphi^{(2)}$, we design a filter that attenuates directions strongly associated with $\theta$, and apply it to the operator $Q^{(2)}$.

In this work, we design a high-pass filter function $H(L^{(1)})$ based on the eigenvalues and eigenvectors of $L^{(1)}$. We denote a monotonically increasing function in $\lambda$ by $h(\lambda) : [0, 1] \to [0, 1]$, and define the graph filters $H(L^{(1)})$ and $H(L^{(2)})$ as follows,

$$H(L^{(1)}) = \sum_i h(\lambda_i^{(1)}) v_i^{(1)} (v_i^{(1)})^T, \quad H(L^{(2)}) = \sum_i h(\lambda_i^{(2)}) v_i^{(2)} (v_i^{(2)})^T. \tag{15}$$

Then we apply $H(L^{(1)})$ to $Q^{(2)}$ to attenuate components in $Q^{(2)}$ associated with the leading eigenvectors of $L^{(1)}$. The same applies to $H(L^{(2)})$ and $Q^{(1)}$:

$$\tilde{Q}^{(2)} = H(L^{(1)})Q^{(2)}H(L^{(1)}), \quad \tilde{Q}^{(1)} = H(L^{(2)})Q^{(1)}H(L^{(2)}). \tag{16}$$

We refer to the leading eigenvectors of the filtered operators $\tilde{Q}^{(1)}$ and $\tilde{Q}^{(2)}$, as *differential vectors* and denote them by $\delta^{(1)}$ and $\delta^{(2)}$. In Sec. 4, we demonstrate that in contrast to the eigenvectors of the unfiltered operators $Q^{(1)}$ and $Q^{(2)}$, under suitable assumptions, these differential vectors are only associated with class-specific latent variables $\varphi^{(1)}$ and $\varphi^{(2)}$. Algorithm 1 summarizes the procedure for extracting a single distinctive latent variable.

### 3.4 EXTRACTION OF MULTIPLE DISTINCTIVE LATENT VARIABLES AND THE PROPOSED FEATURE SCORE

In many cases, data from two different classes within the same dataset may involve multiple distinctive latent variables. In Algorithm 1, we can guarantee that the leading differential vectors of each class, $\delta_0^{(1)}$ and $\delta_0^{(2)}$, are associated exclusively with class-specific latent variables. Here, we need an additional step to guarantee that the subsequent differential vector is non-redundant and associated only with class-specific latent variables that have not yet been identified. We propose an iterative method that updates the filters for the operators $Q^{(1)}, Q^{(2)}$ based on the differential vectors that have been identified. At each iteration, the new graphs for $X^{(1)}$ and $X^{(2)}$ are constructed whose eigenvectors are associated with the shared latent variables, the identified class-specific latent variables, and all their cross-products. Algorithm 2 summarizes the procedure for extracting the multiple distinctive latent variables.

---

**Algorithm 2** Extracting Multiple Distinctive Latent Variables and Proposing a Feature Score

---

1: **Input:** Two datasets $X^{(1)} \in \mathbb{R}^{n_1 \times d}$ and $X^{(2)} \in \mathbb{R}^{n_2 \times d}$, containing $d$ features with $n_1$ and $n_2$ samples respectively. Number of iterations $N$. Filter function $h(\lambda) : [0,1] \rightarrow [0,1]$. The dimension of shared latent space $k_0$.

2: **Output:** Differential vectors $\Delta_0^{(1)}, \ldots, \Delta_{N-1}^{(1)} \in \mathbb{R}^d$. Class-specific feature scores $S_1, S_2$. Feature score $S$.

3: Compute $\delta_0^{(1)}$ as $\Delta_0^{(1)}$ via Algorithm 1.

4: Compute $Q^\theta$ using Eq. 10 and its leading eigenvectors $V^{(0)} \in \mathbb{R}^{d \times k_0}$.

5: **for** $i = 1$ **to** $N - 1$ **do**

6:     Concatenate $V^{(i)} \leftarrow \left[ V^{(i-1)}, \Delta_{i-1}^{(1)} \right]$.

7:     Compute $\Delta_i^{(1)}$ via Algorithm 1 with inputs $X^{(1)}$ and $V^{(i)}$.

8: **end for**

9: Compute the class-specific feature scores $S_1, S_2$ using Eq. 17.

10: Compute the feature score $S$ using Eq. 18.

---

**Proposed Feature Score.** Each differential vector extracted by Algorithm 2 is $d$-dimensional, consistent with the feature dimension of two datasets $X^{(1)}, X^{(2)}$. Consider the differential vectors $\Delta_{N-1}^{(1)}, \Delta_{N-1}^{(2)}$ of $X^{(1)}, X^{(2)}$ obtained in the $N$-th iteration, we denote the class-specific feature scores for the two datasets as $S_1$ and $S_2$, respectively,

$$S_1 = \Delta_{N-1}^{(1)} \odot \Delta_{N-1}^{(1)}, \quad S_2 = \Delta_{N-1}^{(2)} \odot \Delta_{N-1}^{(2)}, \tag{17}$$

where $\odot$ denotes the Hadamard product. Finally, we integrate them to obtain the overall feature score, denoted by

$$S = \max(S_1, S_2), \tag{18}$$

where the maximum is taken element-wise. The proposed feature score $S$ effectively reflects the contribution of each feature in distinguishing the differential structure within the feature space of $X^{(1)}$ and $X^{(2)}$, thereby enabling the identification of features with high discriminative capability.

## 4 THEORETICAL FOUNDATION

We begin with the convergence results of the eigenvectors of the discrete Laplacian to the eigenfunctions of the Laplace-Beltrami (LB) operator under the manifold setting. Theorem 2 in Appendix F.1 implies that the eigenvectors of the random walk Laplacian converge to the eigenfunctions of the LB operator at a certain rate. Based on this theorem, we derive a similar bound for the eigenvectors of the symmetric normalized Laplacian. See the Theorem 3 in Appendix F.1 and the proof in Appendix F.2. Theorems 2 and 3 demonstrate the convergence of the eigenvectors of the graph Laplacian under the manifold corresponding to a single dataset. However, we consider the case of two datasets in our work. In the following, we show that the similar convergence result holds under the product manifold setting associated with two datasets.

**Theorem 1.** *Let $X^{(1)}, X^{(2)}$ be two datasets of $d$ observations sampled uniformly at random from the product manifolds $\mathcal{M}_1 = \mathcal{M}_a \times \mathcal{M}_s, \mathcal{M}_2 = \mathcal{M}_b \times \mathcal{M}_s$ respectively. Let*

$$\delta^{(2)} = \underset{\|v\|=1}{\arg\max} \, v^T P^{(1)} Q_\tau^{(2)} P^{(1)} v \tag{19}$$

*be the differential vector obtained in step 5 of Algorithm 1. Under the assumptions (i)-(iii) for both manifolds, as $d \rightarrow \infty$, with probability at least $1 - 4m^2 d^{-10} - (2m+6)d^{-9}$, the following holds*

$$\left\| \delta^{(2)} - \frac{\alpha}{\sqrt{pd}} \beta_{\pi_b(X)}(f_1^{(b)}) \right\|^2 \leq \mathcal{O}(m\sigma_d) + \mathcal{O}\left( m\sigma_d^{-n/4-1/2} \sqrt{\log d / d} \right). \tag{20}$$

The proof is provided in Appendix F.6. Theorem 1 provides a convergence guarantee for Algorithm 1. It demonstrates that given a sufficiently large number of features, the leading differential vector $\delta^{(2)}$ of the filtered operator $\tilde{Q}^{(2)}$ obtained in step 5 of Algorithm 1 converges to the leading eigenfunction of $\mathcal{M}_b$. Thus, the differential vector captures the leading class-specific latent variable that is not shared between the two datasets.

## 5 EXPERIMENTS

We evaluate the performance of ELVES in comparison with both commonly used and state-of-the-art feature selection methods on a range of synthetic and real-world datasets. In all experiments, the data are split into training and testing sets, and a nested cross-validation strategy is adopted to ensure fair and reliable evaluation. All competing FS methods are carefully tuned to achieve their best performance on the training set. Additional details and results are provided in Appendix B.

### 5.1 MADELON

We evaluate ELVES on the Madelon dataset (Guyon et al., 2008) from the NIPS 2003 feature selection challenge. This dataset contains 2600 samples based on a 5-dimensional hypercube embedded in a 500-dimensional space. Each sample has 500 features, including the 5 hypercube coordinates and 15 of their linear combinations, and the remaining 480 features consist of independent Gaussian noise.

The data is partitioned into train and test sets using 10-fold cross-validation. To assess performance in few-sample scenarios, we consider two settings: (i) the full train set of 2340 samples, (ii) only 5 percent of the train set, comprising 117 samples. Evaluation in both settings is carried out by training an SVM classifier on the full train set and assessing its performance.

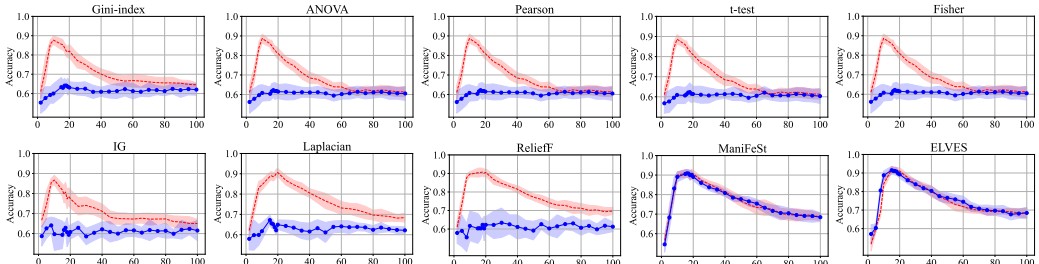

Figure 1: Test accuracy as a function of the feature number on the Madelon dataset. Lines denote the average test accuracy, and the shaded area denote the standard deviation. The red dashed line denotes feature selection using the full training set, while the blue solid line uses only 5% of the data.

Figure 1 shows that when feature selection is performed using the full train set, all methods are able to identify relevant features. It is worth noting that selecting too few or too many features can negatively impact classification performance. ELVES achieves the highest average test accuracy of 91.92%, outperforming ManiFeSt (90.84%) and ReliefF (90.73%). When feature selection is conducted using only 5% of the train set, only ELVES and ManiFeSt identify the relevant features, while all other competing methods fail. In this scenario, ELVES again achieves the best performance with a test accuracy of 91.50%, exceeding ManiFeSt's 90.58%. These results demonstrate that ELVES exhibits remarkable robustness to limited sample sizes and strong generalization capability.

To further demonstrate the robustness of ELVES under few-sample scenarios, we evaluate its performance across varying train set sizes. Figure 2 illustrates the relationship between the maximum classification accuracy and the number of training samples on a logarithmic scale. As shown, all methods except ELVES and ManiFeSt experience a significant performance drop as the sample size decreases, and they completely fail to identify relevant features when the training size falls below 10% (234 samples). In contrast, ELVES exhibits strong robustness to the sample size, maintaining competitive performance even when trained on only 1% of the data (23 samples).

### 5.2 LUNG

We evaluate the advantage of ELVES in extracting class-specific latent variables on the Lung dataset, which contains 203 samples, 3312 features, and 5 classes. We first isolate the data from classes 1 and 2 and perform feature selection separately using ELVES and ManiFeSt. A 10-fold cross-validation procedure is then repeated five times, and the features selected in each iteration are recorded. The average classification accuracies obtained with the top 10, 50, 100, 150, 200, 300 and 500 selected

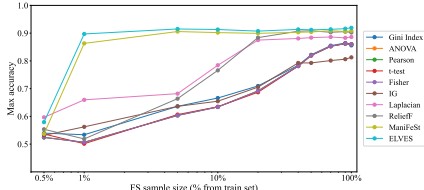

Figure 2: Max test accuracy as a function of the FS sample size on the Madelon dataset.

features are reported in Figure 3(a). Compared with ManiFeSt, ELVES not only produces global feature scores for classes 1 and 2 but also, based on the extracted class-specific latent variables, yields class-specific feature scores for each class (see Eq. 17). This allows ELVES to identify features that are unique to each class. Leveraging this property, when new class data are introduced in practice, previously selected global and class-specific features can be reused. To verify the reusability of the feature selection results produced by ELVES, we consider the case in which data from class 3 of the Lung dataset are added.

For ELVES, we compare the following scenarios: (i) performing feature selection anew on the combined data from all three classes and then training the model, (ii) training the model on the combined data using the previously selected global features from classes 1 and 2, (iii) training the model using the previously selected class-specific features of class 1, and (iv) training the model using the previously selected class-specific features of class 2. The results are shown in Figure 3(c). We observe that, after reusing the feature-selection results from classes 1 and 2 in scenarios (ii)–(iv), the classification accuracies remain very close to those obtained in scenario (i), and in fact scenario (ii) yields slightly higher accuracies than scenario (i) when 200, 300, or 500 features are used. For ManiFeSt, we conduct the same comparison between scenarios (i) and (ii), and the results are shown in Figure 3(b). It can be observed that the performance in scenario (ii) drops substantially compared to scenario (i), further demonstrating that the feature selection results produced by ELVES exhibit markedly superior reusability compared to those of ManiFeSt.

To further demonstrate the discriminative power of the class-specific features identified by ELVES when new class data are introduced, we consider two additional scenarios: (v) performing feature selection and model training anew using data from class 1 (2) together with class 3, and (vi) training the model on the combined class 1 (2) and class 3 data using the previously selected class-specific features of class 1 (2). The results are presented in Figure 3(d) and Figure 3(e), respectively. As shown, the accuracies in the two scenarios are very close, and in some cases almost overlap, indicating that the class-specific features selected by ELVES possess strong discriminative capability.

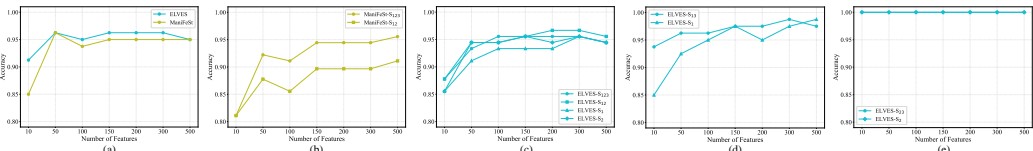

Figure 3: Accuracies under different settings on the Lung dataset. ManiFeSt-$S_{123}$ and ELVES-$S_{123}$ correspond to scenario (i), ManiFeSt-$S_{12}$ and ELVES-$S_{12}$ correspond to scenario (ii), ELVES-$S_1$ corresponds to scenario (iii), ELVES-$S_2$ corresponds to scenario (iv), and so on for subsequent scenarios.

## 5.3 COLON CANCER

We evaluate ELVES on a colon cancer gene expression dataset (Alon et al., 1999), which is widely used in bioinformatics. The dataset comprises 62 tissue samples, each with expression levels measured for 2000 genes. Among them, 22 samples are from normal tissues and 40 from colon cancer tissues. The data are partitioned into 90% for training and 10% for testing, and the results are averaged over 50 cross-validation iterations.

Figure 4 presents the average classification accuracy across different feature subsets for various feature selection methods. ELVES achieves the highest test accuracy of 87.65%, outperforming all competing methods, the best of which attains a lower accuracy of 86.51%. Notably, ELVES demonstrates superior generalization capability compared to all competing methods, as evidenced by the smaller gap between its validation and test accuracies.

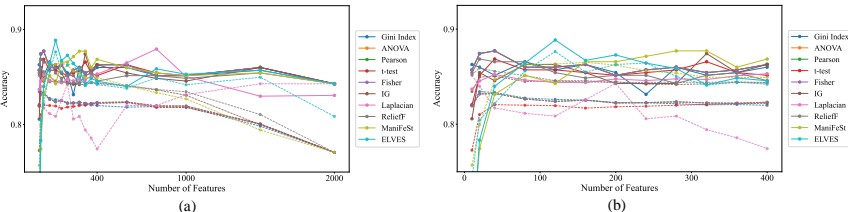

(a)              (b)

Figure 4: Accuracy as a function of the feature number on the colon cancer gene expression dataset. The dashed and solid lines represent the average test and validation accuracy, respectively.

### 5.4 ADDITIONAL RESULTS

We further compare ELVES with different FS methods, including filter-based approaches and the embedded method E2E-FS, on seven additional datasets, all of which exhibit pronounced high-dimensional, few-sample characteristics. Among these, the Lymphoma dataset contains 9 classes, and CLL_SUB_111 contains 3 classes, while the remaining datasets are binary classification problems. Detailed dataset descriptions are in Appendix B. We average the performance obtained with the first 10, 50, 100, 150, and 200 selected features. The results are shown in Table 1. We see that ELVES obtains the highest classification accuracies for all datasets.

Table 1: Comparison of accuracy (%) and standard deviation for various FS methods on benchmark datasets.

| Method | Prostate (5966/102) | ALLAML (7129/72) | Arcene (10000/200) | SMK_CAN_187 (19993/187) | GLI_85 (22283/85) | Lymphoma (4026/96) | CLL_SUB_111 (11340/111) |
|---|---|---|---|---|---|---|---|
| No FS | $91.82 \pm 0.00$ | $97.50 \pm 0.00$ | $83.00 \pm 0.00$ | $77.89 \pm 0.00$ | $84.44 \pm 0.00$ | $94.00 \pm 0.00$ | $68.33 \pm 0.00$ |
| Gini Index | $93.18 \pm 0.64$ | $96.00 \pm 3.35$ | $74.40 \pm 4.83$ | $75.37 \pm 2.31$ | $84.89 \pm 0.99$ | $79.20 \pm 21.34$ | $68.67 \pm 2.47$ |
| ANOVA | $93.63 \pm 1.07$ | $97.50 \pm 3.06$ | $66.80 \pm 3.96$ | $76.42 \pm 2.94$ | $86.22 \pm 0.99$ | $90.80 \pm 7.01$ | $53.67 \pm 3.98$ |
| Pearson | $93.63 \pm 1.07$ | $97.50 \pm 3.06$ | $66.80 \pm 3.96$ | $76.42 \pm 2.94$ | $86.22 \pm 0.99$ | $91.20 \pm 7.56$ | $54.00 \pm 4.80$ |
| t-test | $93.33 \pm 0.75$ | $95.50 \pm 3.26$ | $65.80 \pm 6.38$ | $74.95 \pm 2.02$ | $85.33 \pm 0.99$ | N/A | N/A |
| Fisher | $93.63 \pm 1.07$ | $97.50 \pm 3.06$ | $66.80 \pm 3.96$ | $76.42 \pm 2.94$ | $86.22 \pm 0.99$ | $90.80 \pm 7.01$ | $53.67 \pm 3.98$ |
| IG | $93.21 \pm 0.81$ | $97.50 \pm 3.54$ | $73.40 \pm 5.55$ | $71.37 \pm 3.60$ | $87.56 \pm 2.53$ | $94.40 \pm 3.58$ | $70.21 \pm 3.56$ |
| Laplacian | $91.52 \pm 1.39$ | $91.00 \pm 2.24$ | $69.80 \pm 4.97$ | $72.84 \pm 3.52$ | $71.56 \pm 6.55$ | $92.00 \pm 8.12$ | $57.33 \pm 7.87$ |
| ReliefF | $92.24 \pm 0.73$ | $97.50 \pm 3.06$ | $67.40 \pm 7.57$ | $77.47 \pm 5.35$ | $87.11 \pm 2.43$ | $94.40 \pm 3.85$ | $59.67 \pm 3.98$ |
| Inf-FS | $93.45 \pm 0.92$ | $97.50 \pm 3.06$ | $71.25 \pm 7.50$ | $78.95 \pm 2.27$ | $86.51 \pm 2.48$ | $89.20 \pm 14.46$ | $68.96 \pm 3.55$ |
| ILFS | $92.78 \pm 0.88$ | $94.00 \pm 8.22$ | $69.25 \pm 6.14$ | $79.63 \pm 2.88$ | $88.42 \pm 3.26$ | $83.60 \pm 12.76$ | $70.58 \pm 2.54$ |
| E2E-FS | $93.85 \pm 0.82$ | $88.50 \pm 7.20$ | $71.8 \pm 6.69$ | $76.11 \pm 3.90$ | $76.44 \pm 3.37$ | $83.20 \pm 13.97$ | $61.67 \pm 4.41$ |
| ManiFeSt | $94.10 \pm 0.96$ | $95.00 \pm 3.06$ | $75.40 \pm 4.77$ | $80.23 \pm 3.86$ | $88.89 \pm 3.11$ | $88.00 \pm 9.52$ | $72.36 \pm 2.16$ |
| **ELVES (ours)** | $\mathbf{95.72 \pm 0.56}$ | $\mathbf{98.50 \pm 2.24}$ | $\mathbf{80.00 \pm 7.91}$ | $\mathbf{83.37 \pm 3.34}$ | $\mathbf{90.15 \pm 2.67}$ | $\mathbf{94.60 \pm 4.78}$ | $\mathbf{75.89 \pm 2.64}$ |

*Note:* N/A indicates that the corresponding method is not applicable to multiclass datasets. The first row shows results when all features are used.

## 6 CONCLUSION

In this work, we propose a novel supervised feature selection method which is termed ELVES. ELVES integrates product manifold constructs with spectral graph analysis for feature-space manifold learning, explicitly modeling multivariate feature interactions and capturing class-specific latent variables representing underlying differential structure. We provide an asymptotic convergence analysis demonstrating its ability to reliably discover intrinsic difference patterns. Through comprehensive experiments on diverse benchmark datasets, we demonstrate that our method consistently outperforms state-of-the-art baselines, especially enhancing the generalization performance and exhibiting strong robustness under high-dimensional and few-sample settings.

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

---

**Algorithm 3** Multi-class ELVES Extension

---

1: **Input:** $m$ datasets $\{X^{(i)} \in \mathbb{R}^{n_i \times d}\}_{i=1}^m$, each containing $d$ features with $n_i$ samples in the $i$-th dataset. Number of iterations $N$. Filter function $h(\lambda) : [0,1] \to [0,1]$. The dimension of shared latent space $k_0$.
2: **Output:** Class-specific feature scores $\{S_1, S_2, \ldots, S_m\}$. Feature score $S$.
3: **for** $i = 1$ **to** $m$ **do**
4:     Construct two subsets: $X^{(i)}$ and $X_{\text{rest}}^{(i)} = \bigcup\limits_{\substack{j=1 \\ j \neq i}}^m X^{(j)}$.
5:     Compute the class-specific feature score $S_i$ via Algorithm 2 with inputs $X^{(i)}$ and $X_{\text{rest}}^{(i)}$.
6: **end for**
7: Compute the feature score $S$ using Eq. 22.

---

# A   MULTI-CLASS ELVES EXTENSION

In the method of our paper, we concentrate on binary classification problems, as they serve as the standard stepping-stone to multi-class feature selection (Izetta et al., 2017). In fact, our method can be naturally extended to multi-class problems. Here, we introduce two strategies for extending our method to the multi-class setting. Both approaches follow the same core steps as outlined in Algorithm 2.

The first strategy aims to identify a shared structure operator among multiple classes. As shown in Eq. 10 in Section 3.2 of the paper, we define a shared structure operator for two classes. Under the assumption of product manifolds, this formulation can be generalized to the multi-class case. Specifically, for a classification problem with $m$ classes, we can construct a shared structure operator denoted by

$$Q^\theta = \sum_{1 \leq i < j \leq m} Q^{(i)} Q^{(j)} + Q^{(j)} Q^{(i)}, \tag{21}$$

which captures the underlying structure common to all classes. Assuming that the data from the $m$ classes satisfy the product manifold setting pairwise, then according to Eq. 14, $Q^{(i)} Q^{(j)} + Q^{(j)} Q^{(i)}$ captures the shared structure between class $i$ and class $j$. Consequently, $Q^\theta$ in Eq. 21 captures the shared structure across all pairwise combinations of class data. Based on this shared structure, we compute the class-specific score for each class by pairing the data from the $i$-th class with the shared structure operator $Q^\theta$ as input to Algorithm 2. This process is repeated for all $m$ classes, yielding a set of class-specific scores $\{S_1, S_2, \ldots, S_m\}$. Finally, we aggregate these scores to obtain the overall feature importance score, denoted by

$$S = \max(S_1, S_2, \ldots, S_m), \tag{22}$$

where the maximum is taken element-wise. This strategy allows our method to capture both shared and class-specific structures in the data, making it well-suited for multi-class problems. However, a limitation of this strategy is that the computational cost increases significantly with the number of classes. To address this issue, we propose a second strategy.

The second strategy addresses this issue by adopting a one-vs-rest approach. Specifically, for each class $i$, we divide the dataset into two parts: the samples belonging to class $i$, and those belonging to all other classes. These two subsets are then used as input to Algorithm 2, following the same iterative procedure to compute a class-specific score $S_i$ that captures the discriminative structure of class $i$ against the rest. By repeating this process for all $m$ classes, we obtain class-specific scores $\{S_1, S_2, \ldots, S_m\}$, which are then aggregated according to Eq. 22. This strategy is more computationally efficient and scales better with the number of classes, while still preserving class-specific information.

We adopt the second strategy for extending our method to the multi-class setting. Algorithm 3 summarizes the procedure for the multi-class extension of ELVES.

## B  Experiments - Implementation Details

In all experiments, we adopt a nested cross-validation strategy, where the training set is further split using 10-fold cross-validation to optimize model performance. To ensure robustness on small datasets, the cross-validation process is repeated with randomly shuffled samples. All procedures, including data normalization, feature selection, and SVM hyperparameter tuning, are strictly confined to the training folds to avoid any risk of data leakage.

To enhance the generalization capability of the model, SVM hyperparameters are optimized based on validation data. In contrast, feature selection (FS) method hyperparameters are tuned by maximizing accuracy on the training set alone. This decoupled tuning scheme is adopted to prevent joint optimization of SVM and FS hyperparameters, which can lead to overfitting or confounding effects.

FS hyperparameter tuning is carried out with respect to varying feature subset sizes. As shown in Table 1, we search for optimal hyperparameters using the training data separately for each candidate number of features. The configuration that yields the best performance on the test set is reported. Throughout the rest of the paper, we first identify optimal hyperparameter values for each feature count using a predefined grid search. We then select the setting associated with the highest training accuracy and keep it fixed to evaluate model performance across different feature subsets.

**Data normalization.** Consistent with the approach used by Atashgahi et al. (2022), the features in the Madelon dataset are normalized by centering around the mean and scaling to unit variance. This preprocessing step is carried out using the standard function provided in the sklearn library. No normalization is applied to the remaining datasets, as their feature distributions do not necessitate such treatment.

**FS hyperparameter tuning.** For ReliefF and for IG which extend the classic Information Gain score to continuous features via a $k$ nearest neighbors approach, we tune $k$ over the grid $\{1, 5, 10, 30, 50, 70, 90, 95, 99\}$. For Laplacian Score, which requires a kernel scale parameter, we evaluate scales corresponding to the $\{1, 5, 10, 30, 50, 70, 90, 95, 99\}$ percentiles of the Euclidean distance distribution. ManiFeSt involves only a single scale parameter $\sigma_\ell$. For the illustrative example, $\sigma_\ell$ is set to the median of Euclidean distances. For the XOR and Madelon datasets, due to the pronounced feature interaction patterns, $\sigma_\ell$ is fixed at 0.1 times the median of Euclidean distances without further tuning. For all other datasets, $\sigma_\ell$ is tuned over the $\{5, 10, 30, 50, 70, 90, 95\}$ percentiles, with additional evaluations at the 1st and 99th percentiles to capture nonlocal interactions. The ELVES method involves three key hyperparameters: the number of neighbors $K$ used in constructing the kernel matrix, the number of leading eigenvectors $k$ for the filtering operator, and the number of leading eigenvectors $k_0$ for the shared structure operator. The parameter $K$ is tuned within a range corresponding to 5% to 10% of the total number of features, while $k$ and $k_0$ are selected from a range spanning 90% to 100% of the feature dimension.

**SVM hyperparameter tuning.** When the ground truth regarding feature relevance is unavailable, we evaluate the performance of feature selection using a Support Vector Machine (SVM) with a radial basis function (RBF) kernel. The hyperparameter optimization follows the classical procedure proposed by Hsu et al. (2003), conducting a grid search over exponentially spaced values for both the penalty parameter $C = \{2^{-5}, 2^{-2}, 2^1, 2^4, 2^7, 2^{10}, 2^{13}\}$ and the kernel scale $\gamma = \{2^{-15}, 2^{-12}, 2^{-9}, 2^{-6}, 2^{-3}, 2^0, 2^3\}$.

**Experimental setup.** All experiments were conducted in a Python environment equipped with an RTX 4090 GPU (24GB memory). To accelerate computation, GPU-optimized libraries such as *cupy* and *cuML* were utilized throughout the experiments.

**Implementation of baseline methods.** The implementations of the baseline feature selection methods are as follows: IG and ANOVA were implemented using the *scikit-learn* library. Gini Index, t-test, Fisher Score, Laplacian Score, and ReliefF were implemented using the skfeature package developed by Arizona State University (Li et al., 2017). Pearson correlation was computed using the built-in correlation function from Pandas.

**Details on the hypercube dataset.** In the experiment described in Section C.3, we construct a high-dimensional dataset based on a hypercube embedded in a 10-dimensional space. Specifically, four Gaussian clusters are generated at the vertices of the hypercube, producing a total of 2000 samples. These clusters are arbitrarily grouped into two classes, each containing two clusters, to form a binary

classification task. To increase the dimensionality, the original 10-dimensional data are mapped to a 200-dimensional space by appending random noise to the remaining 190 dimensions. Only the first 10 dimensions carry informative features, while the rest are purely noisy.

**Description of benchmark datasets.** The Prostate cancer dataset contains 5966 gene expression features across 102 samples, with 50 normal and 52 tumor cases. The Gisette dataset derived from the NIPS 2003 feature selection challenge, includes 5000-dimensional feature vectors for 7000 samples. Among these, 2500 dimensions are informative. The classification task is to distinguish between the handwritten digits "4" and "9". RELATHE is a benchmark dataset commonly used in feature selection and dimensionality reduction tasks within the text domain. It is highly sparse, comprising 4322 features and 1427 samples. All experiments adopt a 9:1 train-test split strategy. For the Gisette and RELATHE datasets, results are averaged over 10 cross-validation runs, while for the Prostate cancer dataset, performance is averaged over 30 independent cross-validation iterations.

# C  ADDITIONAL RESULTS

## C.1  TIME AND SPACE COMPLEXITY ANALYSIS OF ELVES

Let $n$ denote the total number of samples, $d$ the number of features, and $N$ the number of extracted differential vectors (typically a small constant such as 5, corresponding to the number of iterations in Algorithm 2). In the following, we provide a theoretical analysis of both the time and space complexity of ELVES.

**Time complexity.** In Algorithm 1, the main costs include computing affinity matrices between features of different classes ($\mathcal{O}(nd^2)$), constructing the graph Laplacian and filtered operators ($\mathcal{O}(d^3)$), and performing spectral decompositions ($\mathcal{O}(d^3)$). In Algorithm 2, this process is repeated $N$ times during the iterative differential vector extraction phase. As a result, the total time complexity is $\mathcal{O}\big(N(nd^2 + d^3)\big)$. This analysis reflects the computational cost associated with modeling the manifold of feature space and extracting class-specific latent variables. Note that when the feature dimension $d$ is very large (e.g., tens of thousands), the eigen-decomposition steps with complexity $\mathcal{O}(d^3)$ may become a computational bottleneck. To address this, approximate methods such as randomized SVD or low-rank kernel approximations (e.g., Nyström method) can be employed in practice.

**Space complexity.** The dominant storage requirements come from kernel matrices, Laplacians, and filtered operators, all of size $d \times d$, as well as a small number of intermediate vectors. Therefore, the overall space complexity is $\mathcal{O}(d^2)$.

## C.2  XOR-100 PROBLEM

To evaluate the capability of ELVES in identifying non-linear feature interactions, we generate a synthetic XOR dataset consisting of $d = 100$ binary features and $N = 50$ samples (Yamada et al., 2020). Each feature is independently drawn from a Bernoulli distribution, and class labels are assigned based on the XOR operation between two predefined features ($f_1$ and $f_5$), making only these two features informative, while the remaining 98 features serve as irrelevant noise. This setup poses a significant challenge to conventional filter methods that rely solely on univariate statistical significance.

We conduct 200 Monte Carlo simulations of data generation. Figure 5 presents the statistical distributions of normalized feature scores for each method. In each simulation, the top two features with the highest scores are selected, and the average of correct selections is shown in parentheses. The results demonstrate that ELVES consistently identifies the interaction between $f_1$ and $f_5$ across all simulations, whereas all other compared methods, except ManiFeSt and ReliefF, have not worked.

Notably, ELVES and ManiFeSt both achieve a perfect average number of correct selections of 2 in identifying the two informative features, with ManiFeSt likewise employing a manifold learning approach in the feature space. Although ReliefF is a univariate method, it indirectly incorporates multivariate information via the local geometry of nearest-neighbor samples and thus also recovers these two features. However, its average of correct selections is only 0.765. This performance gap further underscores the superior ability of ELVES to capture multi-feature interactions.

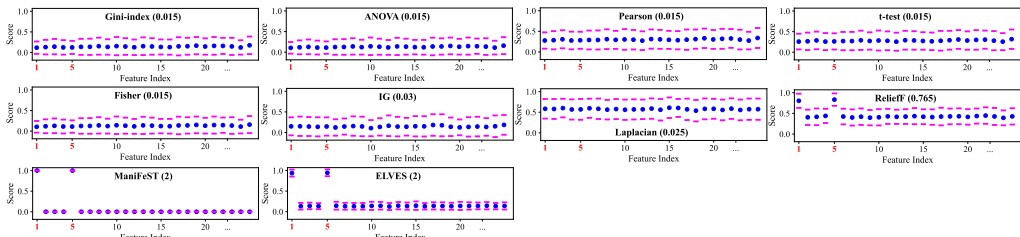

Figure 5: Feature score for the XOR-100 problem. Blue dots denote the average score, and purple lines indicate the standard deviation. The average of correct selections is denoted in parentheses.

### C.3 Clusters on a Hypercube

Due to the absence of ground-truth information on relevant features in the Madelon dataset, we construct a variant of Madelon (Guyon, 2003) using the *make_classification* function from the *scikit-learn* library to enable more accurate evaluation of feature selection performance. In this variant, the ground-truth relevance of features is explicitly known, allowing for direct assessment of feature selection accuracy. More details on the dataset generation are in Appendix B.

The dataset consisting of 200 features is divided into train and test sets with 1500 and 500 samples, respectively. To show the effectiveness of ELVES under few-sample conditions, we perform feature selection using only 50 samples from the train set. For each method, the top 10 ranked features are selected, followed by training an SVM classifier optimized on the full train set. This procedure is repeated for 50 cross-validation iterations.

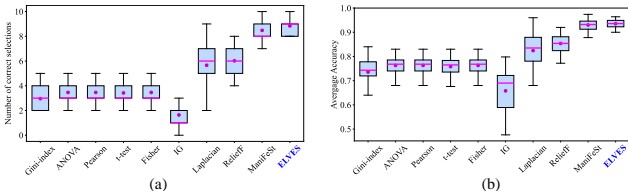

Figure 6: Results on the simulated variant of the Madelon dataset. (a) Boxplot of the number of correct selections. (b) Boxplot of the average accuracy.

Figure 6(a) and 6(b) shows the number of correct selections and the average test accuracy achieved by the evaluated feature selection methods. We see that ELVES attains the highest number of correct selections (8.84) and the highest average test accuracy (0.9352) among all competing methods, surpassing the state-of-the-art approach ManiFeSt, which achieves 8.48 and 0.9301, respectively. These results clearly demonstrate the advantage of ELVES in few-sample scenarios.

### C.4 More Results on additional datasets

We compare ELVES with different FS methods on two additional datasets: Gisette and RELATHE. Dataset descriptions are in Appendix B. The results are shown in Table 2, including the colon cancer dataset. The numbers in parentheses indicate the number of features selected when each method achieved its best performance. As shown, ELVES consistently attains the highest classification accuracies across all datasets.

## D Limitations

Since our method is based on manifold learning in the feature space and primarily accounts for multivariate feature interactions, it may inevitably select some irrelevant features. This limitation highlights a potential direction for future improvement. In addition, the method involves multiple feature decomposition steps, which can lead to substantial computational overhead when dealing

Table 2: Comparison of accuracy (%), standard deviation, and number of features used for various filter FS methods on benchmark datasets.

| Method | Colon (2000/62) | Gisette (5000/7000) | RELATHE (4322/1427) |
|---|---|---|---|
| All features baseline | $82.26 \pm 13.69$ | $97.96 \pm 0.51$ | $74.88 \pm 3.85$ |
| Gini Index | $83.43 \pm 13.38$ (20) | $98.60 \pm 0.50$ (700) | $82.94 \pm 2.97$ (100) |
| ANOVA | $83.23 \pm 12.58$ (40) | $98.65 \pm 0.53$ (800) | $81.68 \pm 3.31$ (100) |
| Pearson | $83.23 \pm 12.58$ (40) | $98.65 \pm 0.53$ (800) | $81.68 \pm 3.31$ (100) |
| t-test | $82.32 \pm 13.83$ (600) | $98.44 \pm 0.47$ (800) | $83.26 \pm 2.93$ (100) |
| Fisher | $83.23 \pm 12.58$ (40) | $98.65 \pm 0.53$ (800) | $81.68 \pm 3.31$ (100) |
| IG | $85.05 \pm 12.53$ (40) | $98.67 \pm 0.52$ (800) | $81.19 \pm 2.50$ (90) |
| Laplacian | $85.14 \pm 12.51$ (18) | $98.60 \pm 0.26$ (1500) | $75.94 \pm 2.49$ (100) |
| ReliefF | $85.92 \pm 12.86$ (20) | $98.67 \pm 0.43$ (1500) | $82.03 \pm 2.82$ (100) |
| ManiFeSt | $86.51 \pm 12.72$ (160) | $98.27 \pm 0.48$ (1000) | $77.48 \pm 4.66$ (100) |
| **ELVES (ours)** | $\mathbf{87.65} \pm 13.91$ (120) | $\mathbf{98.70} \pm 0.45$ (900) | $\mathbf{83.43} \pm 2.11$ (100) |

with very high-dimensional data. However, this cost can be mitigated by leveraging GPU acceleration for matrix decomposition operations.

## E   LARGE LANGUAGE MODEL USAGE STATEMENT

In this work, a large language model is employed to assist in refining the writing of the manuscript. Specifically, the third paragraph of the Introduction and Section 5.2 are polished using an LLM to improve clarity, grammar, and readability. The scientific content, data analysis, and experimental design are entirely generated and verified by ourself.

## F   ADDITIONAL THEORETICAL FOUNDATION

An $l_2$ norm convergence result for the random walk Laplacian $L^{(\mathrm{rw})} = I - D^{-1}K$ was derived in Cheng & Wu (2022) under three assumptions: (i) the $d$ observations are generated uniformly at random over a $n$-dimensional manifold $\mathcal{M}$, (ii) the smallest $m$ eigenvalues of the LB operator over $\mathcal{M}$ have single multiplicity, with a minimal spectral gap $\eta_m > 0$, (iii) the bandwidth parameter of a Gaussian kernel satisfies $\sigma_d \to 0^+$ and $\sigma_d^{n/2+2} > C_m \frac{\log d}{d}$ for some constant $C_m$. Let $\phi_k(X) \in \mathbb{R}^d$ denote the vector of samples given by,

$$\phi_k(X) = \frac{1}{\sqrt{pd}} \beta_X(f_k). \tag{23}$$

where $p$ denotes the uniform sampling density, $f_k$ denotes the normalized eigenfunction corresponding to the $k$-th smallest eigenvalue of the LB operator, and $\beta_X(f_k) \in \mathbb{R}^d$ denotes the sampling operator that evaluates the eigenfunction $f_k$ at a point set $X = \{x_1, \ldots, x_d\} \subseteq \mathcal{M}$. In addition, let $v_k \in \mathbb{R}^d$ and $v_k^{(\mathrm{rw})} \in \mathbb{R}^d$ denote the eigenvectors corresponding to the $k$-th smallest eigenvalues of the symmetric normalized Laplacian and the random walk Laplacian, respectively.

### F.1   THEOREM 2, THEOREM 3 AND LEMMA 4

**Theorem 2** (Theorem 5.5 (Cheng & Wu, 2022)). *Under the assumptions (i)-(iii), for $d \to \infty$, with probability at least $1 - 4m^2 d^{-10} - (4m+6)d^{-9}$, the $k$-th eigenvector of the random walk Laplacian satisfies*

$$\left\| v_k^{(\mathrm{rw})} - \alpha \phi_k(X) \right\|_2 = \mathcal{O}\left( \sigma_d + \sigma_d^{-n/4+1/2}\sqrt{\log d/d} \right), \quad \forall k \leq m \tag{24}$$

*where $v_k^{(\mathrm{rw})}$ is $D$-normalized such that $(v_k^{(\mathrm{rw})})^T D v_k^{(\mathrm{rw})}/(pd) = 1$, $\sigma_d$ is the bandwidth parameter of the Gaussian kernel, and $|\alpha| = 1 + o_p(1)$.*

**Theorem 3.** *Under the assumptions (i)-(iii), for $d \to \infty$, with probability at least $1 - 4m^2 d^{-10} - (4m+8)d^{-9}$, the $k$-th eigenvector of the symmetric normalized Laplacian satisfies*

$$\left\| v_k - \alpha \phi_k(X) \right\|_2 = \mathcal{O}\left( \sigma_d + \sigma_d^{-n/4-1/2}\sqrt{\log d/d} \right), \quad \forall k \leq m \tag{25}$$

*where $\|v_k\| = 1$ is normalized and $|\alpha| = 1 + o_p(1)$.*

The proof of Lemma 3 is in Appendix F.2.

**Lemma 4.** *Let $\mathcal{M}_1 = \mathcal{M}_a \times \mathcal{M}_s$ and $\mathcal{M}_2 = \mathcal{M}_b \times \mathcal{M}_s$. Let $v_{l,k}^{(1)}, v_{m,k'}^{(2)}$ denote the $(l,k)$-th and $(m,k')$-th unit-length eigenvectors of the symmetric normalized Laplacian matrices $L^{(1)}, L^{(2)}$, respectively. We assume that the corresponding eigenvalues $\mu_{l,k}^{(1)}$ and $\mu_{m,k'}^{(2)}$ are both within the $m$ smallest eigenvalues of their respective spectra. Under the assumptions (i)-(iii), for $d \to \infty$, with probability at least $1 - 12m^2 d^{-10} - (8m + 16)d^{-9}$, the inner product between $v_{l,k}^{(1)}$ and $v_{m,k'}^{(2)}$ satisfies*

$$|(v_{l,k}^{(1)})^T v_{m,k'}^{(2)}| = \begin{cases} 1 - \mathcal{O}\left(\sigma_d + \sigma_d^{-n/4-1/2}\sqrt{\log d/d}\right), & \text{if } l = m = 0 \text{ and } k = k'. \\ \mathcal{O}\left(\sigma_d + \sigma_d^{-n/4-1/2}\sqrt{\log d/d}\right), & \text{otherwise.} \end{cases} \quad (26)$$

The proof of Lemma 4 is in Appendix F.5. The lemma plays a crucial role in deriving the convergence guarantee of Algorithm 1. It demonstrates that, apart from the eigenvectors associated only with the shared latent variables $\theta$, the eigenvectors of the Laplacian matrices corresponding to two datasets are nearly orthogonal. Next, we aim to explain why the filtered operator in Eq. 16 is capable of removing the leading eigenvectors associated with the shared latent variables while retaining those associated only with the class-specific latent variables.

Let $M^{(1)} \in \mathbb{R}^{d \times m_1}$ denote the matrix whose columns consist of the eigenvectors of $L^{(1)}$ corresponding to the eigenvalues smaller than the threshold parameter $\tau$ of the filter $H(L^{(1)})$. We define a projection matrix as $P^{(1)} = I - M^{(1)}(M^{(1)})^T$, which is equal to the filter matrix $H(L^{(1)})$. For the analysis, we partition the columns of $M^{(1)}$ to two parts: (i) $M_\theta^{(1)}$ which contains only the eigenvectors associated with the shared latent variables $\theta$, and (ii) $M_\varphi^{(1)}$ which contains the eigenvectors associated with the class-specific latent variables $\varphi^{(1)}$. Then, the following projection matrices can be defined:

$$P_\theta^{(1)} = I - M_\theta^{(1)}(M_\theta^{(1)})^T, \quad P_\varphi^{(1)} = I - M_\varphi^{(1)}(M_\varphi^{(1)})^T. \quad (27)$$

Due to the orthogonality of $M_\theta^{(1)}$ and $M_\varphi^{(1)}$, we have

$$P_\theta^{(1)} P_\varphi^{(1)} = I - M^{(1)}(M^{(1)})^T = P^{(1)}. \quad (28)$$

Similarly, let $M_\theta^{(2)}$ denote the matrix whose columns consists of the eigenvectors of $L^{(2)}$ associated with $\theta$. And we denote a projection matrix by $P_\theta^{(2)} = I - M_\theta^{(2)}(M_\theta^{(2)})^T$. In step 4 of Algorithm 1, we apply a low pass filter to $Q^{(2)}$ and obtain $Q_\tau^{(2)}$ given by

$$Q_\tau^{(2)} = \sum_{l,k;\lambda_{l,k}^{(2)} \leq \tau} (1 - \lambda_{l,k}^{(2)}) v_{l,k}^{(2)} (v_{l,k}^{(2)})^T. \quad (29)$$

Thus, the filtered operator computed in step 4 of Algorithm 1 is equal to $\tilde{Q}^{(2)} = P^{(1)} Q_\tau^{(2)} P^{(1)}$. Note that the projection matrix $P_\theta^{(2)}$ is the *ideal filter* that perfectly eliminates the leading eigenvectors associated with the shared latent variables, whereas $P^{(1)}$ serves merely as a *practical filter*. Let $E_1 = P_\theta^{(2)} Q_\tau^{(2)} P_\theta^{(2)}$ and $E_2 = P_\varphi^{(1)} P_\theta^{(1)} Q_\tau^{(2)} P_\theta^{(1)} P_\varphi^{(1)} = \tilde{Q}^{(2)}$ denote the filtered operators obtained by applying the *ideal filter* and the *practical filter*, respectively. We bound the spectral norm of $E_1 - E_2$ in Appendix F.6.

### F.2 PROOF OF THEOREM 3

Let $L$ be the symmetric normalized Laplacian and $L^{(\mathrm{rw})}$ the random walk Laplacian. The eigenvectors of $L$ satisfy $v_k \propto D^{1/2} v_k^{(\mathrm{rw})}$, where $D$ is the diagonal matrix of degrees and $v_k^{(\mathrm{rw})}$ is the corresponding eigenvector of $L^{(\mathrm{rw})}$ (Von Luxburg, 2007). We take $v_k^{(\mathrm{rw})}$ $D$-normalized such that $(v_k^{(\mathrm{rw})})^T D v_k^{(\mathrm{rw})}/pd = 1$ as in Theorem 2 and $v_k = \frac{D^{1/2}}{\sqrt{pd}} v_k^{(\mathrm{rw})}$ so that $\|v_k\|^2 = \frac{(v_k^{(\mathrm{rw})})^T D v_k^{(\mathrm{rw})}}{pd} = 1$.

By the triangle inequality, we have

$$\left\| v_k - \alpha\phi_k(X) \right\| = \left\| \frac{D^{1/2}}{\sqrt{pd}} v_k^{(\mathrm{rw})} - \alpha\phi_k(X) \right\|$$

$$= \left\| \frac{D^{1/2}}{\sqrt{pd}} v_k^{(\mathrm{rw})} - v_k^{(\mathrm{rw})} + v_k^{(\mathrm{rw})} - \alpha\phi_k(X) \right\|$$

$$\leq \left\| \frac{D^{1/2}}{\sqrt{pd}} v_k^{(\mathrm{rw})} - v_k^{(\mathrm{rw})} \right\| + \left\| v_k^{(\mathrm{rw})} - \alpha\phi_k(X) \right\|. \tag{30}$$

Then we bound the two terms in the right-hand side separately. According to Theorem 2 (Cheng & Wu, 2022, Theorem 5.5), the second term is bounded with probability at least $1 - 4m^2 d^{-10} - (4m+6)d^{-9}$ as

$$\left\| v_k^{(\mathrm{rw})} - \alpha\phi_k(X) \right\| = \mathcal{O}\left( \sigma_d + \sigma_d^{-n/4+1/2} \sqrt{\log d/d} \right). \tag{31}$$

To bound the first term, we need an additional bound on the degree matrix. According to the lemma (Cheng & Wu, 2022, Lemma 3.5), for large values of $d$, with probability at least $1 - 2d^{-9}$, the following holds uniformly for all $i$:

$$D_{ii}/d = c_0 p + \mathcal{O}\left( \sigma_d + \sigma_d^{-n/4} \sqrt{\log d/d} \right), \tag{32}$$

where $c_0$ is a constant determined by the choice of kernel and $p$ is the uniform sampling density on the manifold. For the Gaussian kernel $c_0 = 1$, so we get

$$D_{ii}/pd = 1 + \mathcal{O}\left( \sigma_d + \sigma_d^{-n/4} \sqrt{\log d/d} \right). \tag{33}$$

Then we take the square root of both sides and use a first order expansion of $\sqrt{1+x}$. Thus, we have

$$\sqrt{D_{ii}/pd} = 1 + \mathcal{O}\left( \sigma_d + \sigma_d^{-n/4} \sqrt{\log d/d} \right). \tag{34}$$

Next, we bound the first term of the right-hand side of Eq. 30,

$$\left\| \frac{D^{1/2}}{\sqrt{pd}} v_k^{(\mathrm{rw})} - v_k^{(\mathrm{rw})} \right\| = \left\| \left( \frac{D^{1/2}}{\sqrt{pd}} - I \right) v_k^{(\mathrm{rw})} \right\| \leq \left\| \left( \frac{D^{1/2}}{\sqrt{pd}} - I \right) \right\| \| v_k^{(\mathrm{rw})} \|, \tag{35}$$

where the first term in the right-hand side of Eq. 35 denotes the operator norm. We know that the operator norm of a diagonal matrix is the maximum absolute value of the diagonal elements, given by

$$\left\| \left( \frac{D^{1/2}}{\sqrt{pd}} - I \right) \right\| = \max_i \left| \frac{D_{ii}^{1/2}}{\sqrt{pd}} - 1 \right| = \mathcal{O}\left( \sigma_d + \sigma_d^{-n/4} \sqrt{\log d/d} \right), \tag{36}$$

where this bound holds uniformly for all $i$ with probability at least $1 - 2d^{-9}$. Then we derive a bound for $\| v_k^{(\mathrm{rw})} \|$. Note that $v_k^{(\mathrm{rw})}$ is $D$-normalized such that $(v_k^{(\mathrm{rw})})^T D v_k^{(\mathrm{rw})}/pd = 1$, hence

$$\frac{pd}{\max_i D_{ii}} \leq \| v_k^{(\mathrm{rw})} \|^2 \leq \frac{pd}{\min_i D_{ii}}. \tag{37}$$

By combining Eq. 33 with the first order expansion of $1/(1+x)$, we obtain that both the lower and upper bounds in Eq. 37 are $1 + \mathcal{O}\left( \sigma_d + \sigma_d^{-n/4} \sqrt{\log d/d} \right)$. Thus, by a first order expansion of $\sqrt{1+x}$, we have

$$\| v_k^{(\mathrm{rw})} \| = \sqrt{\| v_k^{(\mathrm{rw})} \|^2} = 1 + \mathcal{O}\left( \sigma_d + \sigma_d^{-n/4} \sqrt{\log d/d} \right). \tag{38}$$

Plugging Eq. 38 and Eq. 36 back into Eq. 35 yields

$$\left\| \frac{D^{1/2}}{\sqrt{pd}} v_k^{(\mathrm{rw})} - v_k^{(\mathrm{rw})} \right\| = \mathcal{O}\left( \sigma_d + \sigma_d^{-n/4} \sqrt{\log d/d} \right). \tag{39}$$

Finally, plugging Eq. 31 and Eq. 39 back into Eq. 30 and applying the union bound over the events where either bound may fail, we conclude that, with probability at least

$$1 - [4m^2 d^{-10} + (4m+6)d^{-9}] - 2d^{-9} = 1 - 4m^2 d^{-10} - (4m+8)d^{-9}, \tag{40}$$

the following bound holds

$$\| v_k - \alpha\phi_k(X) \| = \mathcal{O}\left( \sigma_d + \sigma_d^{-n/4-1/2} \sqrt{\log d/d} \right) \qquad \forall k \leq m. \tag{41}$$

### F.3 Auxiliary lemmas for the proof of lemma 4 and Theorem 1

**Lemma 5.** *Let $v^{(1)} = u^{(1)} - \sigma^{(1)}$ and $v^{(2)} = u^{(2)} - \sigma^{(2)}$ be two vectors that satisfy:*

1. *$\|v^{(1)}\| = \|v^{(2)}\| = 1$.*

2. *$\|\sigma^{(1)}\|, \|\sigma^{(2)}\| = \mathcal{O}(\sigma_d)$.*

3. *The vector $u^{(1)}$ is proportional to $u^{(2)}$ such that $u^{(1)} = cu^{(2)}$ for some constant $c$.*

*Then $|(v^{(1)})^T v^{(2)}| = 1 - \mathcal{O}(\sigma_d)$.*

**Lemma 6.** *Let $U, V \in \mathbb{R}^{d \times m}$ be orthogonal matrices with columns $u_i$ and $v_i$, respectively. Assume that for some $\varepsilon > 0$,*

$$u_i^T v_i \geq 1 - \varepsilon \qquad \forall i = 1, \ldots, m. \tag{42}$$

*Then $\|U - V\|^2 \leq 2m\varepsilon$.*

**Lemma 7.** *Let $M$ be a symmetric positive semi-definite matrix. Let $U, V \in \mathbb{R}^{d \times m}$ be two orthogonal matrices such that $U^T U = V^T V = I$, and let $P_1 = I - UU^T$ and $P_2 = I - VV^T$ be two projection matrices. Then,*

$$\|P_1 M P_1 - P_2 M P_2\| \leq 4\|M\|\|U - V\|. \tag{43}$$

**Lemma 8.** *Let $A \in \mathbb{R}^{d \times d}$ be a symmetric positive semi-definite matrix with spectral decomposition $A = \sum_k \lambda_k u_k u_k^T$, where $u_k, \lambda_k$ are eigenvectors and eigenvalues respectively. Let $V \in \mathbb{R}^{d \times m}$ be a matrix whose columns $\{v_i\}_{i=1}^m$ are orthogonal. Suppose that $|v_i^T u_j| \leq \sigma$ holds for all $(i, j)$, where $\sigma \geq 1/\sqrt{d}$. Then*

$$\|A - (I - VV^T)A(I - VV^T)\| \leq m\sigma^2 \sum_{k=1}^{d} \lambda_k. \tag{44}$$

### F.4 Proofs of auxiliary lemmas

***Proof of Lemma 5.*** We first use the reverse triangle inequality to derive a bound over $u^{(1)}$ and $u^{(2)}$,

$$\|u^{(1)}\| - \|\sigma^{(1)}\| \leq \|v^{(1)}\| = \|u^{(1)} - \sigma^{(1)}\| \leq \|u^{(1)}\| + \|\sigma^{(1)}\|. \tag{45}$$

Thus, combined with assumptions (1) and (2), we get

$$1 + \mathcal{O}(\sigma_d) \geq \|u^{(1)}\| \geq 1 - \mathcal{O}(\sigma_d). \tag{46}$$

A similar bound can be derived for $\|u^{(2)}\|$. Next, we derive a bound for $|(u^{(1)})^T u^{(2)}|$. Since $u^{(1)} = cu^{(2)}$, then $\frac{|(u^{(1)})^T u^{(2)}|}{\|u^{(1)}\|\|u^{(2)}\|} = 1$. Thus, combining Eq. 46 and the above, we have

$$|(u^{(1)})^T u^{(2)}| = \|u^{(1)}\| \cdot \|u^{(2)}\| \geq 1 - \mathcal{O}(\sigma_d). \tag{47}$$

Finally, we derive a bound for $|(v^{(1)})^T v^{(2)}|$. By the reverse triangle inequality, we have

$$|(v^{(1)})^T v^{(2)}| = |(u^{(1)} - \sigma^{(1)})^T (u^{(2)} - \sigma^{(2)})|$$
$$\geq |(u^{(1)})^T u^{(2)}| - |(\sigma^{(1)})^T u^{(2)}| - |(\sigma^{(2)})^T u^{(1)}| - |(\sigma^{(1)})^T \sigma^{(2)}|. \tag{48}$$

According to Eq. 47, the first term is lower bounded by $1 - \mathcal{O}(\sigma_d)$. By the assumption (2), the fourth term is bounded by $\mathcal{O}(\sigma_d^2)$. Using Eq. 46, the second and third terms can be bounded by Cauchy-Schwarz inequality as follows,

$$|(\sigma^{(1)})^T u^{(2)}| \leq \|\sigma^{(1)}\| \cdot \|u^{(2)}\| = \mathcal{O}(\sigma_d). \tag{49}$$

Thus, we demonstrate that

$$|(v^{(1)})^T v^{(2)}| \geq 1 - \mathcal{O}(\sigma_d). \tag{50}$$

**Proof of Lemma 6.** Since $u_i^T v_i \geq 1 - \varepsilon$, we have

$$\|u_i - v_i\|^2 = \|u_i\|^2 + \|v_i\|^2 - 2u_i^T v_i = 2(1 - u_i^T v_i) \leq 2\varepsilon. \tag{51}$$

The spectral norm of a matrix is bounded by its Frobenius norm. Thus,

$$\|U - V\|^2 \leq \|U - V\|_F^2 = \sum_{i=1}^m \|u_i - v_i\|^2 \leq 2m\varepsilon. \tag{52}$$

**Proof of Lemma 7.** Since $M$ is symmetric, $P_1 M P_1 - P_2 M P_2$ is also symmetric. Thus the spectral norm is equal to the largest eigenvalue, given by

$$\|P_1 M P_1 - P_2 M P_2\| = \max_{\|x\|=1} \left| x^T (P_1 M P_1 - P_2 M P_2) x \right|. \tag{53}$$

As $M$ is positive semi-definite, it has a square root, denoted by $M^{1/2}$. Thus, for any vector $x$ we have

$$
\begin{aligned}
\left| x^T (P_1 M P_1 - P_2 M P_2) x \right| &= \left| x^T P_1 M P_1 x - x^T P_2 M P_2 x \right| \\
&= \left| \|M^{1/2} P_1 x\|^2 - \|M^{1/2} P_2 x\|^2 \right| \\
&= \left| \|M^{1/2} P_1 x\| + \|M^{1/2} P_2 x\| \right| \cdot \left| \|M^{1/2} P_1 x\| - \|M^{1/2} P_2 x\| \right|.
\end{aligned} \tag{54}
$$

By Cauchy-Schwarz inequality we have $\|M^{1/2} P_1 x\| \leq \|M^{1/2}\|$ and $\|M^{1/2} P_2 x\| \leq \|M^{1/2}\|$. Thus,

$$\left| \|M^{1/2} P_1 x\| + \|M^{1/2} P_2 x\| \right| \leq 2\|M^{1/2}\|. \tag{55}$$

By the reverse triangle inequality, we have

$$
\begin{aligned}
\left| \|M^{1/2} P_1 x\| - \|M^{1/2} P_2 x\| \right| &\leq \left\| M^{1/2} (P_1 - P_2) x \right\| \\
&\leq \|M^{1/2}\| \|P_1 - P_2\| = \|M^{1/2}\| \|UU^T - VV^T\|.
\end{aligned} \tag{56}
$$

We apply the reverse triangle inequality again to bound the norm $\|UU^T - VV^T\|$. For any vector $x$ we have

$$
\begin{aligned}
\left| x^T (UU^T - VV^T) x \right| &= \left| x^T UU^T x - x^T VV^T x \right| = \left| \|U^T x\|^2 - \|V^T x\|^2 \right| \\
&= \left| \|U^T x\| + \|V^T x\| \right| \cdot \left| \|U^T x\| - \|V^T x\| \right| \\
&\leq 2 \left\| (U - V)^T x \right\| \leq 2\|U - V\|.
\end{aligned} \tag{57}
$$

Thus,

$$\|UU^T - VV^T\| = \max_{\|x\|=1} \left| x^T (UU^T - VV^T) x \right| \leq 2\|U - V\|. \tag{58}$$

Combining the bounds in Eqs. 54, 55, 56 and 58 yields

$$\|P_1 M P_1 - P_2 M P_2\| \leq 2\|M\| \cdot \|UU^T - VV^T\| \leq 4\|M\| \|U - V\|. \tag{59}$$

**Proof of Lemma 8.** Since $A$ is symmetric, $A - (I - VV^T) A (I - VV^T)$ is also symmetric. Thus the spectral norm is equal to the largest eigenvalue, given by

$$\|A - (I - VV^T) A (I - VV^T)\| = \max_{\|z\|=1} \left| z^T \big( A - (I - VV^T) A (I - VV^T) \big) z \right|. \tag{60}$$

First, we prove that the maximizer $z^*$ of Eq. 60 is a linear combination of the vectors in $\{v_i\}_{i=1}^m$ such that $z^* = \sum_{j=1}^m \alpha_j v_j$. Any vector $z$ orthogonal to $V$ satisfies,

$$z^T \big( A - (I - VV^T) A (I - VV^T) \big) z = z^T A z - z^T A z = 0. \tag{61}$$

Thus, the matrix $\big( A - (I - VV^T) A (I - VV^T) \big)$ has a zero eigenvalue with multiplicity $d - m$. Since it is a symmetric matrix, it has exactly $m$ eigenvectors corresponding to non-zero eigenvalues,

and these eigenvectors are contained within the span of $\{v_i\}_{i=1}^m$. This implies that the leading eigenvector $z^*$ is a linear combination of $\{v_i\}_{i=1}^m$. Thus, we have $VV^T z^* = z^*$, and hence

$$(z^*)^T \big(A - (I - VV^T)A(I - VV^T)\big)z^* = (z^*)^T Az^*. \tag{62}$$

Next, we derive a bound over $|v_j^T Av_i|$ for every pair of vectors $v_i, v_j$. Since $|v_i^T u_j| \leq \sigma$, we have

$$|v_j^T Av_i| = \Big|v_j^T \sum_{k=1}^d \lambda_k u_k u_k^T v_i\Big| \leq \sum_{k=1}^d \lambda_k |v_j^T u_k||v_i^T u_k| \leq \sigma^2 \sum_{k=1}^d \lambda_k. \tag{63}$$

Let $z^* = \sum_{i=1}^m \alpha_i v_i$. By Eqs. 62 and 63, we show that the maximal eigenvalue is bounded by

$$|(z^*)^T Az^*| = \Big|\sum_{ij} \alpha_i \alpha_j v_i^T Av_j\Big| \leq \sum_{ij} |\alpha_i \alpha_j v_i^T Av_j| \leq \sum_{k=1}^d \lambda_k \sigma^2 \sum_{ij} |\alpha_i \alpha_j|. \tag{64}$$

Finally, by Cauchy-Schwarz inequality we have that $\sum_{ij} |\alpha_i \alpha_j| = \big(\sum_i |\alpha_i|\big)^2 \leq m \sum_i \alpha_i^2$. Under the unit norm constraint $\sum_i \alpha_i^2 = 1$, the maximum value of $\sum_i |\alpha_i|$ is obtained for $\alpha_i = 1/\sqrt{m}$. Thus, $\sum_{ij} |\alpha_i \alpha_j| \leq m$. Combining Eq. 64 and the above completes the proof.

### F.5 Proof of Lemma 4

By Theorem 3, for all eigenfunctions $f_{l,k}^{(1)}$ corresponding to eigenvalues $\mu_{l,k}^{(1)}$ that are among the smallest $m$ eigenvalues of the Laplace-Beltrami operator on $\mathcal{M}^{(1)}$, with probability at least $1 - 4m^2 d^{-10} - (4m + 8)d^{-9}$ there exists a constant $\alpha^{(1)}$ such that the unit-normalized eigenvector $v_{l,k}^{(1)}$ satisfies

$$\Big\|\frac{\alpha^{(1)}}{\sqrt{p^{(1)}d}}\beta_X(f_{l,k}^{(1)}) - v_{l,k}^{(1)}\Big\|_2 = \mathcal{O}\Big(\sigma_d + \sigma_d^{-n/4 - 1/2}\sqrt{\log d/d}\Big). \tag{65}$$

The same result holds for $\alpha^{(2)}, f_{m,k'}^{(2)}, \mu_{m,k'}^{(2)}$ and $v_{m,k'}^{(2)}$. We denote the differences by

$$\sigma_{l,k}^{(1)} = \frac{\alpha^{(1)}}{\sqrt{p^{(1)}d}}\beta_X(f_{l,k}^{(1)}) - v_{l,k}^{(1)}, \quad \sigma_{m,k'}^{(2)} = \frac{\alpha^{(2)}}{\sqrt{p^{(2)}d}}\beta_X(f_{m,k'}^{(2)}) - v_{m,k'}^{(2)}. \tag{66}$$

To bound the inner product of $v_{l,k}^{(1)}$ and $v_{m,k'}^{(2)}$ we apply the triangle inequality,

$$
\begin{aligned}
|(v_{l,k}^{(1)})^T v_{m,k'}^{(2)}| &= \Big|\Big(\frac{\alpha^{(1)}}{\sqrt{p^{(1)}d}}\beta_X(f_{l,k}^{(1)}) - \sigma_{l,k}^{(1)}\Big)^T\Big(\frac{\alpha^{(2)}}{\sqrt{p^{(2)}d}}\beta_X(f_{m,k'}^{(2)}) - \sigma_{m,k'}^{(2)}\Big)\Big| \\
&\leq \frac{\alpha^{(1)}\alpha^{(2)}}{d\sqrt{p^{(1)}p^{(2)}}}|\beta_X(f_{l,k}^{(1)})^T \beta_X(f_{m,k'}^{(2)})| + \frac{\alpha^{(2)}}{\sqrt{p^{(2)}d}}|(\sigma_{l,k}^{(1)})^T \beta_X(f_{m,k'}^{(2)})| \\
&\quad + \frac{\alpha^{(1)}}{\sqrt{p^{(1)}d}}|(\beta_X(f_{l,k}^{(1)})^T \sigma_{m,k'}^{(2)}| + |(\sigma_{l,k}^{(1)})^T \sigma_{m,k'}^{(2)}|.
\end{aligned}
\tag{67}
$$

Let us address each of these terms separately. The fourth term of Eq. 67 is bounded by

$$|(\sigma_{l,k}^{(1)})^T \sigma_{m,k'}^{(2)}| \leq \|\sigma_{l,k}^{(1)}\|\|\sigma_{m,k'}^{(2)}\| = \mathcal{O}\Big(\Big(\sigma_d + \sigma_d^{-n/4 - 1/2}\sqrt{\log d/d}\Big)^2\Big). \tag{68}$$

Thus, it is negligible. The second and third terms in Eq. 67 can be bounded via the Cauchy-Schwarz inequality. For example, the second term is bounded by

$$\frac{\alpha^{(2)}}{\sqrt{p^{(2)}d}}|(\sigma_{l,k}^{(1)})^T \beta_X(f_{m,k'}^{(2)})| \leq \frac{\alpha^{(2)}}{\sqrt{p^{(2)}d}}\|\sigma_{l,k}^{(1)}\|\|\beta_X(f_{m,k'}^{(2)})\|. \tag{69}$$

By Lemma 3.4 in Cheng & Wu (2022), with probability at least $1 - 2m^2 d^{-10}$, the term $\frac{1}{p^{(2)}d}\|\beta_X(f_{m,k'}^{(2)})\|^2$ is $1 + \mathcal{O}_p(\log d/d)$. Thus, by a first order expansion of $\sqrt{1+x}$, we have

$$\frac{1}{\sqrt{p^{(2)}d}}\|\beta_X(f_{m,k'}^{(2)})\| = 1 + \mathcal{O}_p(\log d/d). \tag{70}$$

Combining this result with the bounds on $\|\sigma_{l,k}^{(1)}\|, \|\sigma_{m,k'}^{(2)}\|$ in Theorem 3, we have

$$\frac{\alpha^{(2)}}{\sqrt{p^{(2)}d}}|(\sigma_{l,k}^{(1)})^T \beta_X(f_{m,k'}^{(2)})| + \frac{\alpha^{(1)}}{\sqrt{p^{(1)}d}}|(\beta_X(f_{l,k}^{(1)}))^T \sigma_{m,k'}^{(2)}| = \mathcal{O}\left(\sigma_d + \sigma_d^{-n/4-1/2}\sqrt{\log d/d}\right). \tag{71}$$

We now bound the first term of Eq. 67. It is equal to,

$$\frac{\alpha^{(1)}\alpha^{(2)}}{d\sqrt{p^{(1)}p^{(2)}}}\left|\beta_X(f_{l,k}^{(1)})^T \beta_X(f_{m,k'}^{(2)})\right|$$

$$= \frac{\alpha^{(1)}\alpha^{(2)}}{d\sqrt{p^{(1)}p^{(2)}}}\left|\left(\beta_{\pi_a(X)}(f_l^{(a)}) \cdot \beta_{\pi_s(X)}(f_k^{(s)})\right)^T \left(\beta_{\pi_b(X)}(f_m^{(b)}) \cdot \beta_{\pi_s(X)}(f_{k'}^{(s)})\right)\right|$$

$$= \frac{\alpha^{(1)}\alpha^{(2)}}{d\sqrt{p^{(1)}p^{(2)}}}\left|\sum_{i=1}^{d} f_l^{(a)}(\pi_a(x_i))f_m^{(b)}(\pi_b(x_i))f_k^{(s)}(\pi_s(x_i))f_{k'}^{(s)}(\pi_s(x_i))\right|. \tag{72}$$

Consider the summands in Eq. 72. Since the coordinates on different manifolds are sampled independently in our setting, the expectation of the summands factorizes,

$$\mathbb{E}[f_l^{(a)}(\pi_a(x))f_m^{(b)}(\pi_b(x))f_k^{(s)}(\pi_s(x))f_{k'}^{(s)}(\pi_s(x))]$$

$$= \mathbb{E}[f_l^{(a)}(\pi_a(x))]\mathbb{E}[f_m^{(b)}(\pi_b(x))]\mathbb{E}[f_k^{(s)}(\pi_s(x))f_{k'}^{(s)}(\pi_s(x))]. \tag{73}$$

By the orthogonality of eigenfunctions with different eigenvalues, we have

$$\mathbb{E}[f_a^{(a)}(\pi_a(x))] = 0 \quad \forall l > 0,$$
$$\mathbb{E}[f_m^{(b)}(\pi_b(x))] = 0 \quad \forall m > 0, \tag{74}$$
$$\mathbb{E}[f_k^{(s)}(\pi_s(x))f_{k'}^{(s)}(\pi_s(x))] = 0 \quad \forall(k \neq k').$$

Hence, except in the special case $l = m = 0$ and $k = k'$, each summand is a mean-zero random variable. Moreover, by Cauchy-Schwarz inequality we can bound the second moment,

$$\mathbb{E}[(f_l^{(a)}(\pi_a(x)))^2] = 1 \quad \forall l,$$
$$\mathbb{E}[(f_m^{(b)}(\pi_b(x)))^2] = 1 \quad \forall m, \tag{75}$$
$$\mathbb{E}\left[\left(f_k^{(s)}(\pi_s(x))f_{k'}^{(s)}(\pi_s(x))\right)^2\right] \leq 1 \quad \forall(k,k').$$

By Eqs. 73, 74 and 75, we obtain that unless $l = m = 0$ and $k = k'$, the sum in Eq. 72 is over i.i.d random variables that are centred with variance $\leq 1$. Since the terms are i.i.d, the variance of the sum is bounded by $d$. Thus by Chebyshev's inequality, the sum is $\mathcal{O}_p(\sqrt{d})$. Therefore,

$$\frac{\alpha^{(1)}\alpha^{(2)}}{d\sqrt{p^{(1)}p^{(2)}}}|\beta_X(f_{l,k}^{(1)})^T \beta_X(f_{m,k'}^{(2)})| = \frac{\alpha^{(1)}\alpha^{(2)}}{\sqrt{p^{(1)}p^{(2)}}}\mathcal{O}_p(1/\sqrt{d}) = \mathcal{O}_p(1/\sqrt{d}). \tag{76}$$

Then we compare the relative magnitudes of $\sigma_d$ and $1/\sqrt{d}$. Note that if $\sigma_d \geq 1$ then $1/\sqrt{d} < \sigma_d$ and if $\sigma_d < 1$ then $1/\sqrt{d} < \sigma_d^{-n/4-1/2}\sqrt{\log d/d}$. In both cases, $1/\sqrt{d}$ is negligible compared with the other terms given in Eqs. 68 and 71. Thus for $k \neq k'$, combining the bounds on the four terms and applying the union bound over the events where either bound may fail, we conclude that, with probability at least

$$1 - \left[2\left(4m^2d^{-10} + (4m+8)d^{-9}\right) + 2\left(2m^2d^{-10}\right)\right] = 1 - 12m^2d^{-10} - (8m+16)d^{-9}, \tag{77}$$

the following bound holds

$$(v_{l,k}^{(1)})^T v_{m,k'}^{(2)} = \mathcal{O}\left(\sigma_d + \sigma_d^{-n/4-1/2}\sqrt{\log d/d}\right). \tag{78}$$

The only remaining case to consider is when $l = m = 0$ and $k = k'$. Since $f_0^{(a)}, f_0^{(b)}$ are both constant functions, then by Eq. 12 we have

$$\beta_X(f_{0,k}^{(1)}) \sim \beta_{\pi_a(X)}(f_0^{(a)})\beta_{\pi_s(X)}(f_k^{(s)}) \sim \beta_{\pi_s(X)}f_k^{(s)}. \tag{79}$$

A similar derivation holds for $\beta_X(f_{0,k'}^{(2)})$. Therefore, we show that $\beta_X(f_{0,k}^{(1)}) \sim \beta_X(f_{0,k'}^{(2)})$. Let $u^{(1)} = \frac{\alpha^{(1)}}{\sqrt{p^{(1)}d}}\beta_X(f_{0,k}^{(1)})$ and $u^{(2)} = \frac{\alpha^{(2)}}{\sqrt{p^{(2)}d}}\beta_X(f_{0,k}^{(2)})$. Applying Lemma 5 to $u^{(1)}, u^{(2)}$ together with $\sigma_{0,k}^{(1)}, \sigma_{0,k}^{(2)}$ yields

$$|(v_{0,k}^{(1)})^T v_{0,k}^{(2)}| = 1 - \mathcal{O}(\|\sigma_{0,k}^{(1)}\| + \|\sigma_{0,k}^{(2)}\|). \tag{80}$$

Combining this result with the bounds on $\|\sigma_{0,k}^{(1)}\|, \|\sigma_{0,k}^{(2)}\|$ given in Theorem 3, we obtain that, if $l = m = 0$ and $k = k'$, then

$$|(v_{0,k}^{(1)})^T v_{0,k}^{(2)}| = 1 - \mathcal{O}\left(\sigma_d + \sigma_d^{-n/4-1/2}\sqrt{\log d/d}\right). \tag{81}$$

This completes the proof.

### F.6 PROOF OF THEOREM 1

Note that $M_\theta^{(2)}$ contains the eigenvectors of $Q_\tau^{(2)}$ associated with the shared latent variables $\theta$. Thus, projecting $Q_\tau^{(2)}$ onto the orthogonal complement of its leading eigenvectors $\{v_{0,1}^{(2)}, v_{0,2}^{(2)}, \ldots, v_{0,m}^{(2)}\}$ gives the matrix $E_1 = P_\theta^{(2)} Q_\tau^{(2)} P_\theta^{(2)}$, which eliminates those eigenvectors. Due to the eigenvalue structure of the product manifold $\mathcal{M}_2$ (see Eq. 13), we obtain that for a sufficiently large $m$, the leading eigenvector of $E_1$ is $v_{1,0}^{(2)}$, which is the leading eigenvector that is not associated with $\theta$. Thus, we have

$$\arg\max_{\|v\|=1} v^T E_1 v = v_{1,0}^{(2)}. \tag{82}$$

Theorem 3 then implies that, as $d \to \infty$,

$$\left\| v_{1,0}^{(2)} - \frac{\alpha}{\sqrt{pd}}\beta_X(f_{1,0}^{(b)}) \right\| = \mathcal{O}\left(\sigma_d + \sigma_d^{-n/4-1/2}\sqrt{\log d/d}\right). \tag{83}$$

To control the deviation between $\delta^{(2)}$ and $v_{1,0}^{(2)}$, it suffices to bound the spectral norm $\|E_2 - E_1\|$, where $E_2 = P_\varphi^{(1)} P_\theta^{(1)} Q_\tau^{(2)} P_\theta^{(1)} P_\varphi^{(1)}$. Adding and subtracting $P_\varphi^{(1)} P_\theta^{(2)} Q_\tau^{(2)} P_\theta^{(2)} P_\varphi^{(1)}$ and then applying the triangle inequality yields

$$\|E_2 - E_1\| = \|P_\varphi^{(1)} P_\theta^{(1)} Q_\tau^{(2)} P_\theta^{(1)} P_\varphi^{(1)} - P_\theta^{(2)} Q_\tau^{(2)} P_\theta^{(2)}\|$$
$$\leq \|P_\varphi^{(1)} P_\theta^{(1)} Q_\tau^{(2)} P_\theta^{(1)} P_\varphi^{(1)} - P_\varphi^{(1)} P_\theta^{(2)} Q_\tau^{(2)} P_\theta^{(2)} P_\varphi^{(1)}\|$$
$$+ \|P_\varphi^{(1)} P_\theta^{(2)} Q_\tau^{(2)} P_\theta^{(2)} P_\varphi^{(1)} - P_\theta^{(2)} Q_\tau^{(2)} P_\theta^{(2)}\|. \tag{84}$$

We now bound the first term of the right-hand side of Eq. 84,

$$\|P_\varphi^{(1)} P_\theta^{(1)} Q_\tau^{(2)} P_\theta^{(1)} P_\varphi^{(1)} - P_\varphi^{(1)} P_\theta^{(2)} Q_\tau^{(2)} P_\theta^{(2)} P_\varphi^{(1)}\| = \|P_\varphi^{(1)}(P_\theta^{(1)} Q_\tau^{(2)} P_\theta^{(1)} - P_\theta^{(2)} Q_\tau^{(2)} P_\theta^{(2)})P_\varphi^{(1)}\|$$
$$\leq \|P_\theta^{(1)} Q_\tau^{(2)} P_\theta^{(1)} - P_\theta^{(2)} Q_\tau^{(2)} P_\theta^{(2)}\|$$
$$\leq 4\|Q_\tau^{(2)}\|\|M_\theta^{(1)} - M_\theta^{(2)}\|. \tag{85}$$

Since $P_\varphi^{(1)}$ is a projection matrix, then $\|P_\varphi^{(1)}\| \leq 1$. By the inequality $\|ABC\| \leq \|A\|\|B\|\|C\|$ that holds for all operator norms, we can prove the first inequality of Eq. 85. The second inequality of Eq. 85 is proved in Lemma 7. Then, combining Lemma 4 with Lemma 6, we get

$$\|M_\theta^{(2)} - M_\theta^{(1)}\|^2 = \mathcal{O}_p\left(m\sigma_d + m\sigma_d^{-n/4-1/2}\sqrt{\log d/d}\right). \tag{86}$$

Thus, combining this result with the bound on the eigenvalues of the operator, $\|Q^{(2)}\| \leq 1$, yields

$$\|P_\varphi^{(1)} P_\theta^{(1)} Q_\tau^{(2)} P_\theta^{(1)} P_\varphi^{(1)} - P_\varphi^{(1)} P_\theta^{(2)} Q_\tau^{(2)} P_\theta^{(2)} P_\varphi^{(1)}\|^2 \leq 16\|Q_\tau^{(2)}\|^2\|M_\theta^{(1)} - M_\theta^{(2)}\|^2$$
$$\leq \mathcal{O}(m\sigma_d) + \mathcal{O}\left(m\sqrt{\frac{\log d}{d\sigma_d^{n/2+1}}}\right). \tag{87}$$

Since each of the bounds above holds with probability at least $1 - 2m^2d^{-10}$ or $1 - (2m+6)d^{-9}$, applying the union bound over all events yields that, all the inequalities above hold simultaneously with probability at least

$$1 - \left[2(2m^2d^{-10}) + (2m+6)d^{-9}\right] = 1 - 4m^2d^{-10} - (2m+6)d^{-9}. \tag{88}$$

Next, we bound the second term of the right-hand side of Eq. 84. Applying Lemma 8 with the matrix $P_\theta^{(2)} Q_\tau^{(2)} P_\theta^{(2)}$ and projection matrix $P_\varphi^{(1)}$, we get

$$\|P_\varphi^{(1)} P_\theta^{(2)} Q_\tau^{(2)} P_\theta^{(2)} P_\varphi^{(1)} - P_\theta^{(2)} Q_\tau^{(2)} P_\theta^{(2)}\| \le m\sigma^2 \Big( \sum_{l,k;\lambda_{l,k}^{(1)}<\tau} (1 - \lambda_{l,k}^{(1)}) \Big). \tag{89}$$

The bound $\sigma$ on the inner products between the eigenvectors of $Q_\tau^{(2)}$ and a vector in the span of $P_\theta^{(2)}$, as required in Lemma 8 is given by Lemma 4. Thus, we have

$$\sigma = \mathcal{O}\left(\sigma_d + \sigma_d^{-n/4-1/2}\sqrt{\log d/d}\right). \tag{90}$$

Plugging this back into Eq. 89 yields

$$\|P_\varphi^{(1)} P_\theta^{(2)} Q_\tau^{(2)} P_\theta^{(2)} P_\varphi^{(1)} - P_\theta^{(2)} Q_\tau^{(2)} P_\theta^{(2)}\| = \Big( \sum_{l,k;\lambda_{l,k}^{(1)}<\tau} (1 - \lambda_{l,k}^{(1)}) \Big) \mathcal{O}\left(m\sigma_d + m\sqrt{\frac{\log d}{d\sigma_d^{n/2+1}}}\right)$$

$$= \mathcal{O}(m^2\sigma_d) + \mathcal{O}\left(m^2\sigma_d^{-n/4-1/2}\sqrt{\log d/d}\right). \tag{91}$$

Since the convergence rate in Eq. 87 is slower than Eq. 91, Then the overall convergence rate of $\|E_2 - E_1\|^2$ is

$$\|E_2 - E_1\|^2 = \mathcal{O}(m\sigma_d) + \mathcal{O}\left(m\sigma_d^{-n/4-1/2}\sqrt{\log d/d}\right). \tag{92}$$

At this point, the Davis-Kahan theorem can be applied to the symmetric matrices $E_1$ and $E_2$ and their respective leading eigenvectors $v_{1,0}^{(2)}$ and $\delta^{(2)}$. According to the theorem, given that $(\delta^{(2)})^T v_{1,0}^{(2)} \ge 0$, we have (Yu et al., 2015, Corollary 1)

$$\|\delta^{(2)} - v_{1,0}^{(2)}\|^2 \le \frac{2^{\frac{4}{3}}\|E_2 - E_1\|^2}{\eta_m^2}, \tag{93}$$

where $\eta_m$ is the minimal spectral gap, which is larger than zero under the assumption (ii). Thus, combining Eq. 92 with Eq. 93 yields

$$\|\delta^{(2)} - v_{1,0}^{(2)}\|^2 \le \mathcal{O}(m\sigma_d) + \mathcal{O}\left(m\sigma_d^{-n/4-1/2}\sqrt{\log d/d}\right). \tag{94}$$

By the triangle inequality, we have

$$\|\delta^{(2)} - \frac{\alpha}{\sqrt{pd}}\beta_{\pi_b(x)}(f_1^{(b)})\|^2 = \|\delta^{(2)} - v_{1,0}^{(2)} + v_{1,0}^{(2)} - \frac{\alpha}{\sqrt{pd}}\beta_X(f_{1,0}^{(b)})\|^2$$

$$\le 2\|\delta^{(2)} - v_{1,0}^{(2)}\|^2 + 2\|v_{1,0}^{(2)} - \frac{\alpha}{\sqrt{pd}}\beta_X(f_{1,0}^{(b)})\|^2$$

$$= \mathcal{O}(\|\delta^{(2)} - v_{1,0}^{(2)}\|^2) + \mathcal{O}(\|v_{1,0}^{(2)} - \frac{\alpha}{\sqrt{pd}}\beta_X(f_{1,0}^{(b)})\|^2). \tag{95}$$

Finally, substituting Eq. 83 and Eq. 94 into Eq. 95 and applying the union bound over the events where either bound may fail, we conclude that, with probability at least $1 - 4m^2d^{-10} - (2m+6)d^{-9}$, the following bound holds

$$\|\delta^{(2)} - \frac{\alpha}{\sqrt{pd}}\beta_{\pi_b(x)}(f_1^{(b)})\|^2 \le \mathcal{O}(m\sigma_d) + \mathcal{O}\left(m\sigma_d^{-n/4-1/2}\sqrt{\log d/d}\right). \tag{96}$$

This completes the proof.

