# OpenReview forum: "ELVES: Extraction of Latent Variables with Enhanced Specificity for High-Dimensional Few-Sample Feature Selection"
_ICLR.cc/2026/Conference — ICLR 2026 Conference Withdrawn Submission_

### Official Review · Reviewer_44KG · 2025-10-23

**Soundness:** 3
**Presentation:** 3
**Contribution:** 2
**Rating:** 4
**Confidence:** 3

**Summary:**

The paper introduces a supervised feature selection method (ELVES) designed to model multivariate feature interactions and capture class-specific latent variables, aiming to reduce redundancy and noise in high-dimensional, few-sample settings. It combines product manifold constructs with spectral graph analysis for feature-space manifold learning and provides an asymptotic convergence analysis to support its theoretical foundation. Experiments on multiple benchmark datasets demonstrate the proposed method’s performance advantages and robustness compared to existing approaches.

**Strengths:**

1. The paper is easy to follow, with clear structure and detailed solution approach.
2. The paper presents detailed mathematical development, including a convergence theorem and related proofs.

**Weaknesses:**

1. The experimental datasets are relatively small, which limits the credibility of the conclusions. Large-scale real-world datasets related to vision or language representation is missing.
2. The font size used in the tables is too small, which hinders readability.
3. The authors lack a deeper analysis of the effectiveness of the proposed method and a theoretical comparison with existing approaches.
4. The authors carefully tuned the method to achieve the best performance. However, it remains unclear how the proposed method performs compared to other approaches under fixed parameter settings.

**Questions:**

See weaknesses.

---

### Official Review · Reviewer_g6c7 · 2025-10-28

**Soundness:** 3
**Presentation:** 3
**Contribution:** 3
**Rating:** 4
**Confidence:** 3

**Summary:**

This paper introduces **ELVES**, a supervised feature selection method designed for high-dimensional, few-sample (HDFS) data. Unlike traditional approaches, ELVES leverages the **feature space manifold** by constructing class-specific feature kernels to capture feature interactions. It then applies product manifold theory and spectral graph analysis to design a filter that extracts class-specific latent variables, which are subsequently used to score and identify discriminative features. Experimental results on benchmark datasets demonstrate that ELVES outperforms several state-of-the-art feature selection methods.

**Strengths:**

- The paper targets the practically important HDFS problem, which is challenging and relevant to real-world applications.

- The proposed method achieves superior accuracy and robustness compared to existing feature selection baselines.

- Theoretical analysis is provided to support the convergence of the proposed method.

**Weaknesses:**

1. The proposed framework is conceptually similar to some existing approaches that explore class-specific Laplacian matrices, even though those methods may not explicitly focus on feature selection.

2. The computational complexity could become a practical bottleneck when applied to high-dimensional datasets with limited samples.

**Questions:**

1. Redundant features can distort the manifold structure. How does the proposed method mitigate the influence of redundancy n the learned manifold?

2. The class-specific latent variables may encode both informative and redundant features. How does ELVES distinguish between them?

3. Why is a high-pass filter used for the feature measurement process?

4. The method appears to rely on selecting a certain number of eigenvectors. How sensitive is the performance to this selection, and how is the number determined?

---

### Official Review · Reviewer_mToi · 2025-10-30

**Soundness:** 3
**Presentation:** 2
**Contribution:** 2
**Rating:** 4
**Confidence:** 3

**Summary:**

The paper proposes a supervised feature selection method for classification problems that is based on a hybrid manifold model to identify features that are correlated across classes from features that are independent in each class. The goal is to be able to identify feature sets that capture both data structure that is pervasive and data structure that is discriminative. The approach is based on the construction of graph laplacians to model correlations within a class and across classes, which are then use to perform filtering using a Graph Fourier transform, and exploiting specific structures in those laplacians. The filter outputs are then use to obtain feature scores.

**Strengths:**

The use of hybrid manifold models is novel, and the role served by the graph based analysis is sound. Some experimental results are compelling in the comparison against baselines (e.g., Table 1).

**Weaknesses:**

For the experimental results, it is common to see how the feature selection approaches fare against one another as the number of features is increased. The average performance shown in Table 1 may hide some of the intricacies of each method working better or worse for small vs. large numbers of chosen features. Ideally one method is consistently better than the others, but this does not occur often in practice.

Manifold models are not often motivated by real-world applications; some discussion as to how they may arise for the examples considered would be helpful. In addition, the assumption of a "separable" manifold for each class with a shared component and an independent component in the latent variable space (eq. 1) could be better motivated in practical settings. This is further refined into a "manifold product" assumption (eq. 11) that could be better motivated as well. For example, it would be useful to see specific choices of variables selected for each of the manifold components in the examples. One would expect a large degree of interpretability in feature selection, and this should carry on to the model proposed and its components.

There are multiple instances of undefined notation: in eq. (20), $\beta_{\pi_b}$ has not been defined and is not defined in the proof either (it first appears in eq. (23) but once again undefined).

**Questions:**

Some terminology is not clear. In particular:

In lines 185-186, what determines if an eigenvector is "associated" with a shared latent variable or a class-specific latent variable?

In line 213, how are eigenvalues ordered by a doublet (l,k)?

In line 241, what are "directions strongly associated with $\theta$"?

In Algorithm 2, lines 3 and 7 use Algorithm 1 , which has two dataset inputs and computes two differential vectors, but the inputs and outputs are not described in this way in these two instances.

What is the classification problem used in the Madelon dataset?

Can you elaborate on why the results of Figure 3 are split among five sub-figures? The caption does not address (a-e) individually.

---

### Official Review · Reviewer_mnPs · 2025-11-01

**Soundness:** 3
**Presentation:** 2
**Contribution:** 2
**Rating:** 2
**Confidence:** 3

**Summary:**

This paper proposes ELVES (Extraction of Latent Variables with Enhanced Specificity), a feature selection framework that integrates graph signal processing and product manifold modeling to extract class-specific latent variables, aiming to improve feature relevance and robustness in high-dimensional, few-sample scenarios.

**Strengths:**

Clear Motivation: Considering that feature selection under high-dimensional and small-sample conditions remains a well-recognized challenge, this paper addresses this scenario directly and demonstrates clear application value.

Technical Rigor: The authors combine graph signal processing, spectral graph theory, and the assumption of product manifolds to construct a coherent feature scoring framework, which is rigorously validated on multiple real-world and synthetic datasets.

Comprehensive Experiments: The empirical section includes extensive experiments across diverse datasets and a broad range of baselines. ELVES consistently outperforms competing methods in both accuracy and robustness under few-sample conditions.

**Weaknesses:**

1. The technical components employed in this work (e.g., Gaussian kernels, normalized Laplacians, graph filtering, and product manifold modeling) are all well-established tools. The method primarily integrates existing spectral graph and manifold learning techniques, resulting in limited conceptual novelty.

2. The extraction of discriminative information through one-vs-rest or inter-class filtering is a classical idea. While the “high-pass filtering” design in ELVES may appear novel in formulation, conceptually it does not move beyond traditional discriminative modeling frameworks and lacks comparison with more recent feature selection paradigms, such as those based on mutual information, causal inference, or contrastive learning.

3. The paper lacks a clear overall flowchart or schematic illustration of the proposed framework, which makes it difficult for readers to follow the methodological pipeline and understand how the components of ELVES interact.

**Questions:**

1. The paper defines the high-pass filter function h($\lambda$) as a monotonically increasing function, but its exact form or implementation is not specified. What specific function is used in the experiments, and how sensitive are the results to this choice?

2. Given the high computational complexity, how does ELVES scale to datasets with very high feature dimensionality?

3. The experimental comparison could be strengthened by including more recent feature selection methods, such as those based on mutual information estimation, causal discovery, or contrastive learning, to provide a more comprehensive and up-to-date evaluation.

4. In the multi-class setting, ELVES relies on a one-vs-rest strategy, which could make feature scoring unstable under class imbalance. Have the authors considered this potential issue?

---

### Official Review · Reviewer_VbxV · 2025-11-02

**Soundness:** 3
**Presentation:** 3
**Contribution:** 3
**Rating:** 4
**Confidence:** 4

**Summary:**

This paper proposes ELVES, a novel supervised feature selection method for high-dimensional, few-sample (HDFS) data. The core idea is to model the manifold structure of the feature space to capture inter-feature interactions.

**Strengths:**

Novelty: The core idea of modeling class-specific manifolds in the feature space is highly innovative and effectively captures multivariate feature dependencies.

Theoretical Grounding: The method is supported by a theoretical convergence guarantee, which adds significant rigor.

**Weaknesses:**

Computational Cost: The time complexity of O(d³) makes the method potentially infeasible for extremely high-dimensional datasets without approximations, which were not experimentally validated.
Theoretical Assumptions: The method relies on a product manifold assumption, and the impact of violating this assumption on real-world data is not discussed.
Lack of Intuition: The paper is heavy on mathematical formalism and could benefit from more intuitive explanations for its core mechanisms.
Hyperparameter Sensitivity: The method introduces several key hyperparameters, but the paper lacks a thorough sensitivity analysis or practical guidance for their selection.

**Questions:**

Regarding scalability: How does ELVES perform on datasets with hundreds of thousands of features? What is the performance trade-off when using approximation methods like Nyström or randomized SVD?
Regarding the product manifold assumption: Could you provide more intuition on real-world scenarios where this assumption is likely to hold and how performance degrades when it is violated?
Regarding the filter design: What is the intuition behind choosing a monotonically increasing function h(λ) as the high-pass filter? Have alternative filter designs been explored?

---

### Note · Authors · 2026-01-29

I have read and agree with the venue's withdrawal policy on behalf of myself and my co-authors.

---

### Meta-Review · Area_Chair_abyV · 2026-01-07

**Summary:**

The reviewers agree that the paper addresses an important problem—feature selection in high-dimensional, few-sample regimes—and appreciate the combination of spectral graph analysis, product manifold modeling, and a convergence-style theoretical analysis. Several reviewers find the empirical results strong and competitive across a range of datasets and baselines. However, the majority of reviewers also raise concerns regarding the practical scalability of the method due to high computational complexity, the strength and realism of the modeling assumptions (particularly the product/hybrid manifold assumption), limited conceptual novelty given reliance on established tools, and clarity/correctness issues in the presentation. These concerns collectively prevent strong confidence in acceptance at this stage.

**Reviewer Concerns:**

Concerns acknowledged include the relevance of the problem setting, the inclusion of theoretical analysis, and competitive empirical performance. However, critical concerns remain unresolved. These include scalability and practicality due to O(d³) complexity without validated approximations, insufficient motivation and robustness analysis of the product/hybrid manifold assumptions, limited conceptual novelty due to reliance on established techniques, clarity and correctness issues such as undefined notation and unclear algorithm descriptions, and experimental limitations related to dataset scale and hyperparameter sensitivity.

**Reviewer Scores:**

Overall, the scores cluster around a weak reject, with one stronger reject and no clear advocates for acceptance. The authors also didn't participate in rebuttal.

---

### Decision · Program_Chairs · 2026-01-26

Reject